# Risk factors for healthcare-associated infections in children undergoing ECMO after cardiac surgery for congenital heart disease: A retrospective study

Fanwei Meng[1,2], Bing Chen[1]*

1 Department of Emergency, The Second Hospital of Tianjin Medical University, Tianjin, China,
2 Department of Extracorporeal Circulation and Extracorporeal Life Support, Fuwai Central China Cardiovascular Hospital, Zhengzhou, Henan, China

* cb20202023@163.com

## Abstract

### Objective

To analyze the influencing factors of ECMO (extracorporeal membrane oxygenation)-related healthcare-associated infections in children after congenital heart disease (CHD) surgery.

### Methods

A retrospective analysis was conducted on the healthcare-associated infections (HAI) and related factors of 54 pediatric patients who received ECMO support after CHD surgery at Fuwai Central China Cardiovascular Hospital between 1 January 2019 and 1 April 2024. All children were divided into the infection group (n = 14) and the non-infection group (n = 40) based on whether they developed ECMO-related HAIs. Multivariate logistic regression analysis was used to identify the risk factors for ECMO-related HAI in children after congenital heart disease surgery.

### Results

Among the 54 ECMO-supported children, 14 developed HAIs during ECMO support, with a total of 17 pathogenic microorganisms isolated. Gram-negative bacteria predominated (3 strains of Klebsiella pneumoniae, 3 strains of Stenotrophomonas maltophilia, 2 strains of Pseudomonas aeruginosa, and 2 strains of Burkholderia cepacia), accounting for 64.7% of the cases. Statistically significant differences were observed between the infected and non-infected groups in terms of preoperative endotracheal intubation for congenital heart disease and ICU stay duration ($p < 0.05$), but there was no significant difference in mortality between the two groups ($p > 0.05$). Multivariate logistic regression analysis revealed that preoperative endotracheal intubation for

**Data availability statement:** All relevant data are within the paper and its Supporting information files.

**Funding:** The author(s) received no specific funding for this work.

**Competing interests:** The authors have declared that no competing interests exist.

congenital heart disease (OR = 4.852, 95% CI: 1.174–20.059; $p = 0.029$) was significantly associated with the occurence of ECMO-related HAI.

## Conclusion

ECMO-related HAI in children after congenital heart surgery are primarily Gram-negative bacterial respiratory infections. Preoperative endotracheal intubation for congenital heart disease was significantly associated with the occurence of ECMO-related HAI in children after CHD surgery. Avoiding preoperative endotracheal intubation whenever possible may reduce the incidence of HAI. However, given the observational design and potential for confounding by severity, these findings should be interpreted as hypothesis-generating rather than definitive. Future studies are needed to determine whether strategies to minimize preoperative intubation or optimize infection prevention in intubated patients can reduce HAI rates in this high-risk population.

## Introduction

Children with congenital heart disease who experience low cardiac output syndrome or cannot be weaned from cardiopulmonary bypass after surgery frequently require ECMO support [1–2]. Due to the surgical trauma and underlying conditions, these patients face an elevated risk of acquiring healthcare-associated infections (HAI) during ECMO assistance, which further compromises their prognosis [3]. ECMO patients exhibit a markedly higher incidence rate of HAI compared to non-ECMO patients (27.7% vs. 7.9%) [4]. Extracorporeal Life Support Organization (ELSO) data demonstrates that the occurrence rate of ECMO-related HAIs is 11% ineonates, 21.4% in children, and peaks at 27% in adults [5]. ECMO support therapy is linked to increased medical costs, and elevated HAI rates further exacerbate medical expenses and patient hazards [6]. Consequently, there is a urgent need for effective infection control measures and prevention strategies to reduce HAI rates during ECMO support. However, there are relatively few reports on the influencing factors of ECMO-related HAI in children after congenital heart surgery. This study aims to explore the risk factors and pathogenic microbiology of ECMO-related HAI in pediatric patients following congenital heart surgery, providing a reference for the prevention of HAI.

## Materials and methods

### Data Collection

The clinical data of 54 pediatric patients who received ECMO support after congenital heart surgery at Fuwai Central China Cardiovascular Hospital between 1 January 2019 and 1 April 2024 were retrospectively collected (Fig 1). Data access began on June 1, 2024. Data analysis was conducted from December 1, 2024, to March 1, 2025. The data included general clinical information (age, sex, weight, primary diagnosis) and microbiological laboratory results. Postoperative follow-up was conducted via telephone.

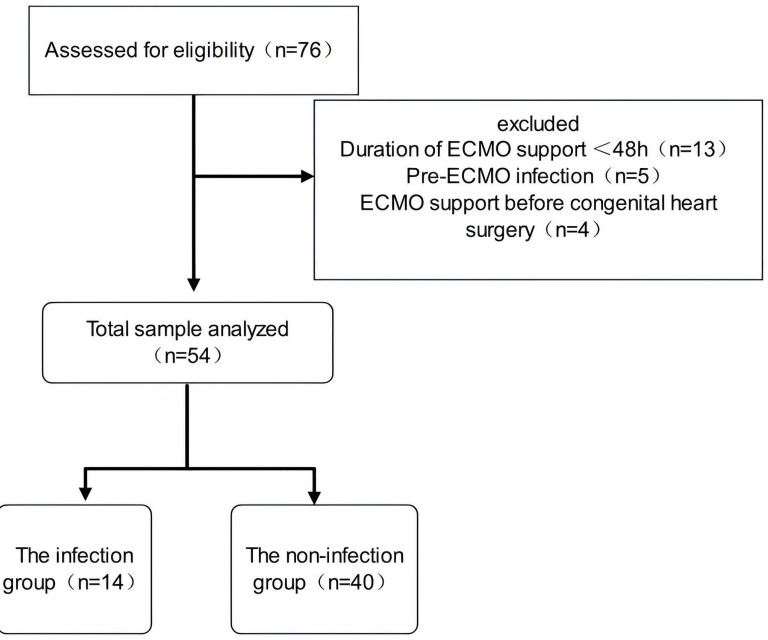

**Fig 1. Flow diagram of patient selection.**

Inclusion Criteria:

1. Age＜18 years;

2. ECMO support duration＞48 hours.

Exclusion Criteria:

1. Presence of healthcare-associated infections before ECMO initiation;

2. Patients who required ECMO support before congenital heart surgery.

All data were collected by two trained researchers from the hospital's electronic medical record system using a standardized data collection form. To ensure patient confidentiality, all personal identifiers (e.g., name, medical record number) were removed and each patient was assigned a unique study code prior to analysis. The anonymized data were stored in a password-protected database, accessible only to the principal investigator.

The ECMO-related HAI was defined as infections occurring between 24 hours after ECMO initiation and 48 hours after ECMO decannulation, with microbiological confirmation through positive cultures (from blood, respiratory tract, surgical sites, wounds, urine, etc) or next-generation sequencing (NGS) detection [4]. Based on the children's chest X-ray findings and laboratory test results, empirical antibiotic therapy was administered when infection was highly suspected, and the antibiotic regimen was adjusted after the results of bacterial culture and drug sensitivity test were obtained. For children with clinically highly suspected infection but negative culture results, we empirically administer antibiotics; however, such children were not included in the infection group in this study. Based on infection status, patients were categorized into either the infection group or non-infection group. This study was approved by the Ethics Committee of Fuwai Central China Cardiovascular Hospital (Approval Number: 2024−24). The requirement for informed consent was waived by the Ethics Committee Union Hospital, Fuwai Central China Cardiovascular Hospital based on the study's retrospective analysis of patient data.

Respiratory tract infections:

1. New onset purulent secretions or change in sputum colour, odour, quantity or consistency; and at least one positive culture with an identifiable pathogen/organism;

2. Radiological evidence of new and persistent infiltrates (two or more serial chest X-rays or CT scans with suggestive images of pneumonia).

Bloodstream infection:

1. At least one positive culture with an identifiable pathogen/organism; OR

2. Patient has one of the following signs or symptoms: fever (>38.0°C), chills or hypotension.

Urinary tract infection(UTI):
two of the following with no other recognised cause:

1. Fever >38.0°C, urgency, frequency, dysuria, suprapubic tenderness;

2. Positive dipstick for leucocyte esterase and/or nitrate, pyuria urine specimen with >10 White Blood Cell (WBC)/mL, organisms seen on Gram stain of unspun urine, at least two urine cultures with repeated isolation of the same uro-pathogen (Gram-negative bacteria or Staphylococcus saprophyticus) [7].

## ECMO Cannulation and Management

All patients received venoarterial ECMO (VA-ECMO) support. Our hospital is a nationally accredited ECMO Quality Control Center which holds official ELSO certification. Nurse-to-patient ratio: 2:1. All children were treated in the ICU and managed jointly by ECMO-qualified doctors, nurses, and perfusionists. For patients unable to be weaned from cardiopulmonary bypass, central cannulation was employed. For postoperative low cardiac output syndrome, femoral arteriovenous cannulation was used in children weighing ≥20 kg, while central cannulation was preferred for those <20 kg. ECMO management followed the Extracorporeal Life Support Organization (ELSO) guidelines. Initial ECMO flow was maintained at 100–150 ml/(kg·min), with gradual reduction of vasoactive drugs based on blood pressure to maintain mean arterial pressure between 30–50 mmHg. Routine microbiological surveillance (including sputum and blood cultures) was performed every 2–3 days. Systemic anticoagulation with heparin was administered, with target activated clotting time (ACT) of 160–220 seconds and activated partial thromboplastin time (APTT) maintained at 1.5–2 times normal values.

## Statistical Analysis

Statistical analysis was conducted using SPSS 25.0. Continuous variables with a normal distribution were expressed as mean ± standard deviation and compared using Student's t-test. Non-normally distributed continuous variables were presented as median with interquartile range [M(Q1,Q3)] and analyzed using the Mann-Whitney U test. Categorical variables were reported as frequencies (percentages) and compared using either chi-square test or Fisher's exact test, as appropriate. Given the very small cell counts for several diagnostic categories in the infected group, these categories were grouped before applying the Chi-square test. These were divided into two groups. Fisher's exact test was used for categorical variables when more than 20% of the expected cell frequencies were less than 5. Multivariate logistic regression analysis was performed to identify independent risk factors for ECMO-related HAIs following congenital heart surgery. Model performance was evaluated using two approaches. First, model calibration which reflects the agreement between observed and predicted outcomes, was assessed using the Hosmer-Lemeshow goodness-of-fit test, where a $p > 0.05$ indicates adequate calibration. Second, model discrimination defined as the ability of the model to distinguish between

patients with and without the outcome, was evaluated by calculating the Area Under the Receiver Operating Characteristic curve (AUC). A two-tailed $p < 0.05$ was considered statistically significant.

## Results

### Microbiological Profile

Among the 14 infected pediatric cases, 17 pathogenic strains were isolated, with the following distribution: Gram-negative bacteria (11 strains, 64.7%): Klebsiella pneumoniae (3 strains); Stenotrophomonas maltophilia (3 strains); Pseudomonas aeruginosa (2 strains); Burkholderia cepacia (2 strains); Acinetobacter baumannii (1 strain); Gram-positive bacteria (6 strains, 35.3%): Staphylococcus epidermidis (2 strains); Enterococcus faecium (1 strain); Staphylococcus aureus (1 strain); Streptococcus mitis (1 strain); Streptococcus pneumoniae (1 strain). Infection Sites: Respiratory tract infections: 14 cases (100%); Secondary bloodstream infections: 3 cases (21.4%) (Table 1).

### Univariate analysis of ECMO-related HAIs in children after congenital heart disease (CHD) surgery

There were no statistically significant differences between the infection group and the non-infection group in terms of gender, age, body weight, admission diagnosis, cardiopulmonary bypass time, and ascending aorta cross-clamping time. There were also no statistically significant differences between the two groups in terms of lactate, blood urea nitrogen, serum creatinine, bilirubin, and albumin before extracorporeal membrane oxygenation (ECMO). There were no statistically significant differences between the two groups in the aspects of the location where ECMO was initiated, postoperative chest drainage volume within 24 hours, the blood transfusion volume 24 hours after ECMO, peritoneal dialysis, continuous renal replacement therapy (CRRT), extracorporeal cardiopulmonary resuscitation (ECPR), the duration of ventilator support before ECMO, and the duration of ECMO assistance ($p > 0.05$). There were statistically significant differences between the infection group and the non-infection group in the length of stay in the intensive care unit (ICU) and preoperative endotracheal intubation in children with congenital heart disease ($p < 0.05$) (Table 2).

### Multivariate logistic regression analysis of risk factors for HAIs in children after congenital heart disease (CHD) surgery

Variables with $p < 0.05$ in the univariate analysis were included in the multivariate logistic regression analysis, which revealed that preoperative endotracheal intubation in children with congenital heart disease (OR = 4.852, 95%

**Table 1. Pathogenic microorganisms of ECMO-associated HAIs in children after congenital heart disease (CHD) surgery (n = 17).**

| Microorganism category | n | Pathogen | Respiratory tract infections | Bloodstream infections |
|---|---|---|---|---|
| Gram-negative bacteria | 11 | Klebsiella pneumoniae | 3 | 0 |
| | | Stenotrophomonas maltophilia | 3 | 0 |
| | | Pseudomonas aeruginosa | 2 | 0 |
| | | Burkholderia cepacia | 2 | 0 |
| | | Acinetobacter baumannii | 1 | 0 |
| Gram-positive bacteria | 6 | Staphylococcus epidermidis | 0 | 2 |
| | | Staphylococcus aureus | 0 | 1 |
| | | Enterococcus faecium | 1 | 0 |
| | | Streptococcus mitis | 1 | 0 |
| | | Streptococcus pneumoniae | 1 | 0 |

**Table 2.** Univariate analysis of ECMO-associated HAIs in children after congenital heart disease (CHD) surgery [$\overline{X} \pm s$, M(Q1, Q3), n(%)].

| Group | The infection group (n = 14) | The non-infection group (n = 40) | $\chi^2$/t | *p-value* |
|---|---|---|---|---|
| Gender (male/female) | 7/7 | 20/20 | 0.001 | 1.000 |
| age (Neonates) | 2 | 10 | | 0.71 |
| body weight [kg] | 8.1±6.7 | 8.6±10.3 | 0.781 | 0.863 |
| **admission diagnosis/n** | | | | 0.448 |
| tetralogy of Fallot | 2(14.3%) | 3(7.5%) | | |
| DORV | 0(0%) | 3(7.5%) | | |
| Mitral Regurgitation | 0(0%) | 4(10.0%) | | |
| PA | 2(14.3%) | 4(10.0%) | | |
| TAPVC | 4(28.6%) | 5(12.5%) | | |
| TGA | 2(14.3%) | 6(18.0%) | | |
| CoA / IAA | 0(0%) | 8(20.0%) | | |
| others | 4(28.6%) | 7(17.5%) | | |
| CPB time (min) | 203.8±118.2 | 243.1±102.0 | 1.035 | 0.239 |
| ascending aorta cross-clamping time (min) | 87.7±42.6 | 113.0±59.3 | 1.351 | 0.162 |
| Blood lactate (mmol/l) | 9.1±5.7 | 11.9±8.6 | 1.979 | 0.264 |
| Postoperative chest drainage volume within 24 hours [ml, M(Q1,Q3)] | 100.0 (26.3, 195.0) | 92.5 (40.0, 256.3) | | 0.789 |
| the blood transfusion volume 24 hours after ECMO (ml) | 260.5±180.8 | 411.4±425.1 | 3.146 | 0.205 |
| BUN (mmol/L) | 11.1±5.1 | 8.2±5.4 | 0.139 | 0.087 |
| serum creatinine (umol/L) | 66.3±43.7 | 66.3±39.5 | 0.536 | 0.998 |
| Bilirubin [umol/L, M(Q1,Q3)] | 33.0 (18.7, 51.3) | 40.0 (19.2, 78.7) | | 0.269 |
| Albumin (g/L) | 36.8±7.5 | 33.0±8.0 | 0.003 | 0.128 |
| The location of ECMO initiation (operating room/ICU) | 7/7 | 23/17 | 0.236 | 0.627 |
| Cannulation approach (central / peripheral) | 11/3 | 38/2 | | 0.103 |
| duration of ventilator support before ECMO (d) | 13.2±10.1 | 8.9±6.5 | | 0.071 |
| the length of stay in the ICU [d, M(Q1,Q3)] | 21.5 (16.3, 48.0) | 16.0 (6.5, 24.5) | | 0.023 |
| the duration of ECMO assistance [d, M(Q1,Q3)] | 8.0 (5.8, 13.3) | 7.0 (4.0, 10.0) | | 0.139 |
| peritoneal dialysis [n(%)] | 9 (64.3%) | 30 (75.0%) | 0.593 | 0.441 |
| CRRT [n(%)] | 2 (14.3%) | 10 (25.0%) | | 0.282 |
| ECPR [n(%)] | 2 (14.3%) | 4 (10.0%) | | 0.643 |
| preoperative endotracheal intubation in children with CHD [n(%)] | 8 (57.1%) | 8 (20.0%) | 6.862 | 0.009 |
| death [n(%)] | 7 (50.0%) | 25 (62.5%) | 0.671 | 0.413 |

Abbreviations: DORV, Double Outlet Right Ventricle; PA, Pulmonary Atresia; TAPVC, Total Anomalous Pulmonary Venous Connection; TGA, Transposition of the Great Arteries; CoA, Coarctation of the Aorta; IAA, Interrupted Aortic Arch; CPB, cardiopulmonary bypass; ECPR, Extracorporeal cardiopulmonary resuscitation; CRRT, continuous renal replacement theray; BUN, blood urea nitrogen; ECMO, Extracorporeal Membrane Oxygenation; CHD, congenital heart disease; ICU, intensive care unit; Lactate, blood urea nitrogen, serum creatinine, bilirubin and albumin were measured before extracorporeal membrane oxygen.

CI: 1.174–20.059; *p*=0.029) was significantly associated with the occurence of ECMO-related HAI in pediatric postcardiotomy patients (Table 3). The Hosmer-Lemeshow goodness-of-fit test yielded a non-significant result ($\chi^2$=10.67, df=8, *p*=0.221), indicating no significant deviation between observed and predicted event rates. Regarding model discrimination, the area under the receiver operating characteristic curve (AUC) was 0.686 (95% CI: 0.514–0.858), suggesting marginal but statistically significant discriminatory ability (Fig 2).

 

**Table 3. Multivariate logistic regression analysis of risk factors for HAIs in children after congenital heart disease (CHD) surgery. Abbreviations:** *B*, regression coefficient; S.E., standard error; Wald, Wald test; OR, odds ratio; CI, confidence interval.

| Variable | B | S.E | Wald | p-value | OR-value | 95%CI |
|---|---|---|---|---|---|---|
| The length of stay in the ICU | 0.025 | 0.019 | 1.716 | 0.190 | 1.026 | 0.988~1.07 |
| Preoperative endotracheal intubation in children with CHD | 1.579 | 0.724 | 4.757 | 0.029 | 4.852 | 1.174~20.059 |

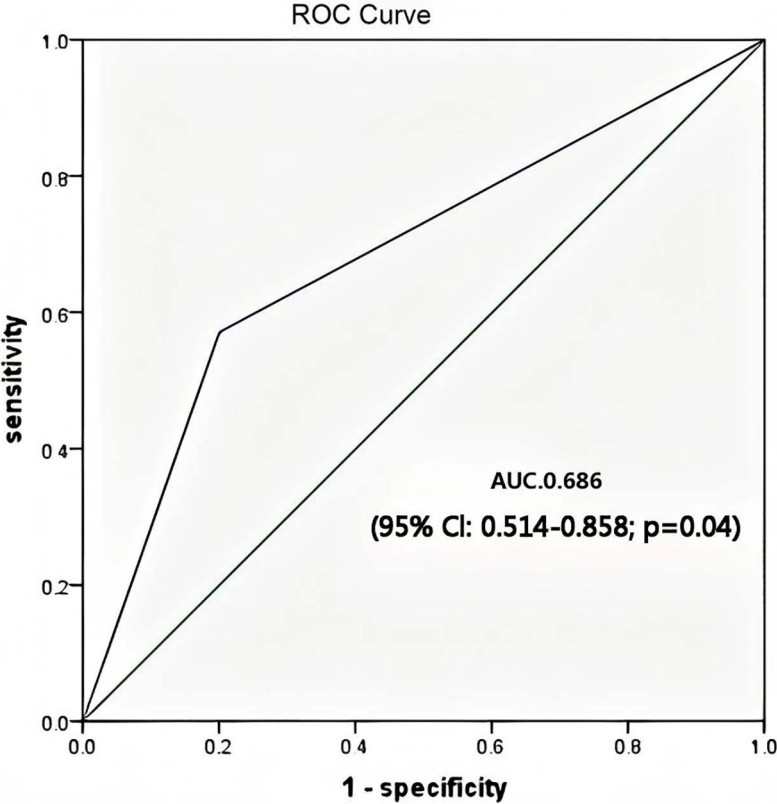

**Fig 2. The ROC curve analysis for the multivariate logistic regression model.**

## Discussion

This study conducted a retrospective analysis of pediatric patients who received ECMO support following congenital heart disease surgery. The incidence of ECMO-related healthcare-associated infection (HAI) in this cohort was 25.9%, which is generally consistent with reported rates from international studies on ECMO-associated HAI in children [8]. During the ECMO support process in children, the diagnosis of HAI is relatively difficult. Due to the presence of the cooling water tank and the systemic inflammatory response caused by the contact between blood and foreign substances (extracorporeal circulation and ECMO-related pipelines), it cannot be clearly diagnosed through clinical signs (fever), laboratory tests (white blood cells, C-reactive protein, procalcitonin) and chest X-ray findings [9]. Therefore, in this study, microbiological culture results were utilized as the primary diagnostic criterion for HAI, an approach that has been widely adopted in previous investigations [10–11]. Previous reports have indicated that ECMO-associated HAIs are predominantly bloodstream infections, followed by respiratory tract infections [12]. In contrast, among the 14 infected children in our cohort, all presented

with respiratory tract infections, while the incidence of bloodstream infection was relatively low (21.4%). Several factors may account for this discrepancy. First, all subjects were post-cardiotomy patients requiring ECMO support; pre-existing intracardiac anomalies (such as left-to-right shunting and pulmonary vein obstruction) predisposed these children to pulmonary congestion, thereby increasing susceptibility to pulmonary infection [13]. Second, routine prophylactic antibiotic administration during ECMO support in our institution may have reduced the positivity rate of blood cultures [14]. Consistent with previous studies identifying Gram-negative bacteria as the predominant pathogens in ECMO-related HAI [15–16], our analysis revealed that Gram-negative organisms accounted for 64.7% of culture-positive cases, predominantly opportunistic pathogens including Klebsiella pneumoniae, Stenotrophomonas maltophilia, and Pseudomonas aeruginosa.

Multivariate logistic regression analysis in this study identified preoperative endotracheal intubation was associated with the occurence of HAI during ECMO support in children after cardiac surgery. A meta-analysis involving over 50,000 pediatric patients found that HAI had the highest incidence among tracheal intubation-related complications, reaching 11.9% [17]. These findings suggest that preoperative intubation may predispose patients to subsequent ECMO-associated HAI by compromising the airway mucosal barrier and increasing pathogen colonization/invasion risk. Potential mechanisms include: (1) direct introduction of pathogens during intubation, with subsequent mechanical ventilation causing mucosal damage and localized immunosuppression; (2) underlying critical illness in intubated congenital heart disease patients (such as cardiac dysfunction or severe hypoxemia) that independently increases HAI susceptibility. While these observations align with existing literature on invasive procedures elevating infection risks in critically ill children [18–19], the present study uniquely highlights the specific contributory role of preoperative intubation within the ECMO context, where prolonged extracorporeal circulation and immune dysregulation may amplify this effect.

Notably, while ECMO-associated HAI did not significantly impact overall survival rates, they markedly prolonged ICU length of stay ($p < 0.05$). This phenomenon may be attributed to infection-related complications (sepsis, organ dysfunction) requiring extended organ support and antibiotic therapy. Early recognition and targeted antimicrobial treatment may have effectively mitigated infection-related mortality [20–21]. However, prolonged ICU stays not only increase healthcare costs but also raise the risk of secondary complications, including catheter-related infections, coagulation disorders, and neurological sequelae [22–24], potentially affecting long-term outcomes and imposing additional financial burdens on families. ICU length of stay includes both preoperative and postoperative ICU stay. If a child is in stable condition before surgery and does not require endotracheal intubation, they will not be admitted to the ICU preoperatively, resulting in a relatively shorter total ICU stay. Conversely, if a child requires endotracheal intubation before surgery, the total ICU stay will be prolonged. Notably, the relationship between ICU stay and infection is complex and bidirectional. Relevant studies have indicated that HAI following congenital heart disease surgery prolong the length of ICU stay, while preoperative ICU admission is one of the risk factors for developing HAI [25]. Given this bidirectional relationship—wherein infection may prolong ICU stay and prolonged ICU stay may increase infection risk—we cannot establish a clear temporal sequence. Therefore, ICU length of stay should be interpreted as a marker of overall clinical complexity rather than a direct risk factor for HAI. The shorter ICU stay observed in the non-infected group may also reflect treatment withdrawal due to disease severity or financial constraints, suggesting a potential bidirectional relationship that warrants further investigation to clarify causal associations. Future studies employing time-dependent analytical approaches are needed to clarify the temporal dynamics between ICU stay and HAI.

Several limitations of this study should be acknowledged.

First, confounding by indication represents a major methodological concern. Preoperative endotracheal intubation is likely a marker of underlying clinical severity rather than a direct causal factor. Children requiring preoperative intubation typically present with more severe cardiac physiology, hemodynamic instability, or preoperative critical illness—factors that may independently increase the risk of healthcare-associated HAI. Consequently, the observed association between intubation and HAI should be interpreted as an association rather than a causal relationship.

Second, our definition of HAI relied on microbiologically confirmed cases, which may introduce selection and misclassification bias. Clinically suspected infections that received antimicrobial treatment but lacked positive culture results were

excluded. This conservative approach likely underestimates the true incidence of ECMO-related HAI. Additionally, prior antimicrobial therapy may have suppressed bacterial growth, leading to false-negative culture results—a common diagnostic challenge in this population.

Third, several statistical limitations warrant consideration. The small sample size and low number of infection events limit the precision and stability of our estimates. Regarding the multivariable model, although the events-per-variable ratio (EPV = 7) met the threshold typically considered adequate for standard logistic regression, the borderline EPV still carries a risk of overfitting. The 95% confidence interval for preoperative endotracheal intubation (OR=4.852, 95% CI: 1.174–20.059) remained wide, indicating considerable estimation imprecision. This imprecision is likely attributable to the limited number of events within this specific subgroup, warranting cautious interpretation of the magnitude of the observed association. The model demonstrated marginal but statistically significant discriminatory ability (AUC 0.686, 95% CI: 0.514–0.858) and adequate calibration (Hosmer-Lemeshow test $p$ = 0.221). However, the predictive power of the model is limited, as the lower bound of the confidence interval approaches 0.5, indicating discrimination only slightly better than random chance. The reported odds ratios should be interpreted with caution in clinical practice, as the model's predictive accuracy for individual patient risk is constrained. Given these limitations, our findings should be interpreted as exploratory and hypothesis-generating rather than confirmatory or predictive.

Fourth, the retrospective design introduces inherent biases, and the single-center nature of the data may limit generalizability. Additionally, the lack of stratified analysis examining pathogen types and antibiotic usage patterns represents a further constraint. Future multicenter prospective studies incorporating microbiological evidence and immune function monitoring are needed to further elucidate the pathological mechanisms of ECMO-associated HAI and optimize prevention strategies.

In summary, healthcare-associated infections (HAI) during ECMO support following pediatric congenital heart surgery were predominantly Gram-negative respiratory infections. Preoperative endotracheal intubation was significantly associated with the occurence of HAI, suggesting that this patient subgroup warrants heightened attention. However, given the observational design and potential for confounding by severity, these findings should be interpreted as hypothesis-generating rather than definitive. Future studies are needed to determine whether strategies to minimize preoperative intubation or optimize infection prevention in intubated patients can reduce HAI rates in this high-risk population.

## Supporting information

**S1 File. Raw data**.
(PDF)

## Author contributions

**Data curation:** Fanwei Meng.

**Formal analysis:** Fanwei Meng, Bing Chen.

**Funding acquisition:** Bing Chen.

**Investigation:** Fanwei Meng.

**Writing – original draft:** Fanwei Meng.

**Writing – review & editing:** Bing Chen.

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
