## [Decision Letter · Decision Letter 0]

18 Sep 2025

PONE-D-25-38491Analysis of Risk Factors for ECMO-Associated Nosocomial Infections in Children After congenital heart disease SurgeryPLOS ONE

Dear Dr. Meng,

Thank you for submitting your manuscript to PLOS ONE. After careful consideration, we feel that it has merit but does not fully meet PLOS ONE’s publication criteria as it currently stands. Therefore, we invite you to submit a revised version of the manuscript that addresses the points raised during the review process.

We look forward to receiving your revised manuscript.

Kind regards,

Giovanni Giordano

Academic Editor

PLOS ONE

Journal Requirements:

- https://doi.org/10.3389/fcvm.2022.1071575

In your revision ensure you cite all your sources (including your own works), and quote or rephrase any duplicated text outside the methods section. Further consideration is dependent on these concerns being addressed.

5. In the online submission form, you indicated that all data included in this study are available upon request by contact with the corresponding author.

6. Please update your submission to use the PLOS LaTeX template. The template and more information on our requirements for LaTeX submissions can be found at http://journals.plos.org/plosone/s/latex.

Reviewers' comments:

Reviewer's Responses to Questions

**Comments to the Author**

1. Is the manuscript technically sound, and do the data support the conclusions?

Reviewer #1: Yes

Reviewer #2: Yes

2. Has the statistical analysis been performed appropriately and rigorously? 

Reviewer #1: Yes

Reviewer #2: Yes

3. Have the authors made all data underlying the findings in their manuscript fully available?

Reviewer #1: Yes

Reviewer #2: Yes

4. Is the manuscript presented in an intelligible fashion and written in standard English?

Reviewer #1: Yes

Reviewer #2: Yes

5. Review Comments to the Author

Reviewer #1: Thank you for allowing me the opportunity to review the manuscript "Analysis of Risk Factors for ECMO-Associated Nosocomial Infections in Children After congenital heart disease Surgery".

My main concern is the definition of infection in the patient population - while positive respiratory cultures generally translates into infection, it could also be colonization. It would be interesting to look at the chest imaging and ECMO settings along with laboratory results to determine if they were real infections or not. On the other hand, the authors should also acknowledge the fact that positive cultures alone may underestimate clinical infections if antibiotics were used prior to cannulation.

I also recommend to discuss potential confounding when it comes to ICU length of stay - a time-to-event analysis could be implemented.

Reviewer #2: Major Comments

The retrospective single-center design with a relatively small cohort (n=54) limits the generalizability of the findings. Please expand the discussion on how the limited sample size may affect the robustness of the logistic regression analysis. The authors rely exclusively on microbiological confirmation, without considering clinical and biochemical markers (e.g., CRP, PCT). This approach could underestimate clinically relevant infections. A comparison with alternative diagnostic criteria would strengthen the manuscript. In univariate analysis, both ICU stay and pre-ECMO ventilation duration showed associations with infection, but they lost significance in multivariate analysis. Please comment on possible interactions between these variables and preoperative intubation. The conclusion emphasizes avoiding preoperative intubation when possible. The authors should provide more practical recommendations (e.g., airway management strategies, antimicrobial prophylaxis protocols, VAP prevention bundles) to enhance clinical applicability.

6. PLOS authors have the option to publish the peer review history of their article (what does this mean?). If published, this will include your full peer review and any attached files.

Reviewer #1: No

Reviewer #2: No

---

## [Author Response · Author response to Decision Letter 1]

1 Oct 2025

Rviewers1

Response: Thank you very much for your constructive feedback.

1、During the ECMO support process in children, the diagnosis of infection is relatively difficult. Due to the presence of the cooling water tank and the systemic inflammatory response caused by the contact between blood and foreign substances (extracorporeal circulation and ECMO-related pipelines), it cannot be clearly diagnosed through clinical signs (fever), laboratory tests (white blood cells, C-reactive protein, procalcitonin) and chest X-ray findings. Pulmonary injuries after cardiac surgery, such as pulmonary edema and atelectasis, can also affect the diagnosis of infection.In this study, only microbial indicators were used as the indicators of infection, and this diagnostic standard has been widely adopted and applied.

2、For pediatric patients with relatively high infection indicators, antibiotics may be administered as appropriate before cannulation. This can lead to negative culture results, resulting in an underestimation of clinical infections.

3、It has been supplemented.

The length of ICU stay includes pre - operative ICU stay and post - operative ICU stay. If a child is in a stable condition before surgery and does not require endotracheal intubation, they will not be transferred to the ICU, resulting in a relatively shorter overall ICU stay. Conversely, if a child needs endotracheal intubation before surgery, the total ICU stay will be prolonged. On the other hand, children who require pre - operative endotracheal intubation are usually in more severe conditions, leading to a longer post - operative recovery period and thus an extended post - operative ICU stay. In other words, pre - operative endotracheal intubation will increase the length of children's ICU stay.

Rviewers 2

Response: Thanks for your key comment! We have carefully addressed your valuable suggestions as follows：

1、The main risks of insufficient sample size (failing to meet the Events Per Variable≥10, EPV) for Logistic regression are the phenomena of complete separation and quasi-complete separation. In terms of results, these risks manifest as either the inability to obtain the regression coefficient B, or an excessively large Wald (resulting in an extremely small p-value). However, in this analysis, the regression coefficient B is 1.601, with a Wald of 4.43 (p=0.035), and neither complete separation nor quasi-complete separation is observed.

It was a retrospective single center study using observationsal analysis. The sample size of this study was small, which does not meet the requirements of events per variable. However, considering the relatively small number of ECMO-Associated Nosocomial Infections in children after congenital heart disease surgery, the results are valuable. However, the reliability of this result needs further confirmation.

2、During the ECMO support process in children, the diagnosis of infection is relatively difficult. Due to the presence of the cooling water tank and the systemic inflammatory response caused by the contact between blood and foreign substances (extracorporeal circulation and ECMO-related pipelines), it cannot be clearly diagnosed through clinical signs (fever), laboratory tests (white blood cells, C-reactive protein, procalcitonin) and chest X-ray findings. In this study, only microbial indicators were used as the indicators of infection, and this diagnostic standard has been widely adopted and applied.For pediatric patients with relatively high infection indicators, antibiotics may be administered as appropriate before cannulation. This can lead to negative culture results, resulting in an underestimation of clinical infections.

3、Pediatric patients requiring preoperative endotracheal intubation are in critical condition, which will increase their post-operative ICU length of stay. The subjects of our study are pediatric patients receiving VA ECMO support after cardiac surgery; preoperative intubation will inevitably lead to prolonged pre-ECMO ventilation duration.

4、The specific recommendations are as follows:

(1)Strict preoperative evaluation of intubation indications, with efforts to minimize the time interval between intubation and surgery.

(2)Meticulous aseptic technique during intubation and intensified postoperative airway management.Use an airway filter to assist with humidification while preventing airway contamination. Strictly adhere to aseptic techniques: wash hands and wear sterile gloves before suctioning, use disposable suction catheters to avoid cross-infection; administer 100% oxygen for 30 seconds prior to suctioning to prevent transient hypoxemia induced by suctioning from exacerbating cardiac burden.

(3)Close monitoring of infection markers during ECMO, with prophylactic antimicrobial strategies for high-risk patients.Antibiotics should only be used for pediatric patients with clear high-risk factors for infection before surgery, such as pre-existing pulmonary infection (chest X-ray indicating inflammation, elevated white blood cell count/CRP in blood routine).For patients without clear high-risk factors, routine prophylactic use of antibiotics before surgery is not recommended; only strict aseptic techniques should be implemented during intubation to reduce the risk of iatrogenic infection.Broad-spectrum antibiotics are preferred, and specific selections should be adjusted based on the hospital's bacterial resistance surveillance data.

(4)Ventilator-Associated Pneumonia (VAP) Prevention Measures: During non-surgical periods, elevate the head of the bed by 30°-45° to reduce reflux and aspiration of gastric contents, thereby lowering the risk of VAP. Perform oral care every 6 hours to inhibit the colonization of pathogenic bacteria in the oral cavity and reduce retrograde airway infection caused by bacteria. Ventilator circuits: Replace every 7 days; if contamination (e.g., sputum contamination) occurs, replace immediately. Drain condensed water in a timely manner to prevent it from refluxing into the airway or humidifier.

---

## [Decision Letter · Decision Letter 1]

7 Nov 2025

PONE-D-25-38491R1Analysis of Risk Factors for ECMO-Associated Nosocomial Infections in Children After congenital heart disease SurgeryPLOS ONE

Dear Dr. Meng,

Thank you for submitting your manuscript to PLOS ONE. After careful consideration, we feel that it has merit but does not fully meet PLOS ONE’s publication criteria as it currently stands. Therefore, we invite you to submit a revised version of the manuscript that addresses the points raised during the review process.

We look forward to receiving your revised manuscript.

Kind regards,

Giovanni Giordano

Academic Editor

PLOS ONE

Journal Requirements:

Reviewers' comments:

Reviewer's Responses to Questions

**Comments to the Author**

1. If the authors have adequately addressed your comments raised in a previous round of review and you feel that this manuscript is now acceptable for publication, you may indicate that here to bypass the “Comments to the Author” section, enter your conflict of interest statement in the “Confidential to Editor” section, and submit your "Accept" recommendation.

Reviewer #2: (No Response)

Reviewer #3: All comments have been addressed

2. Is the manuscript technically sound, and do the data support the conclusions?

Reviewer #2: Yes

Reviewer #3: Partly

3. Has the statistical analysis been performed appropriately and rigorously? 

Reviewer #2: Yes

Reviewer #3: No

4. Have the authors made all data underlying the findings in their manuscript fully available?

Reviewer #2: Yes

Reviewer #3: Yes

5. Is the manuscript presented in an intelligible fashion and written in standard English?

Reviewer #2: Yes

Reviewer #3: Yes

6. Review Comments to the Author

Reviewer #2: The authors have substantially improved the manuscript and adequately addressed all major comments from the previous review. The discussion is now clearer and more clinically grounded, and the addition of practical recommendations increases the translational relevance. Only minor revisions remain.

1. Statistical limitations

The authors added a paragraph on the limitations of small sample size for logistic regression. It would be useful to explicitly report the events-per-variable ratio (EPV) calculated for this model to quantify the degree of statistical limitation.

2. Diagnostic criteria paragraph

The justification for microbiological-only diagnosis is sound, but please clarify whether negative cultures with high clinical suspicion (e.g., prior antibiotics) were recorded or excluded. This will help readers interpret potential underestimation bias.

3. Terminology consistency

Use consistent terminology throughout: “ECMO-associated nosocomial infection” vs. “ECMO-related infection.” Consider harmonizing the terms in title, abstract, and text.

4. Discussion on causality

The manuscript mentions the possible bidirectional relationship between ICU stay and infection. It would strengthen the discussion to acknowledge that reverse causality cannot be excluded in this retrospective design.

Reviewer #3: Dear Authors,

Thank you for the opportunity to review your manuscript investigating infections in pediatric patients undergoing ECMO after congenital heart surgery. The topic is clinically relevant and of great importance for improving patient safety and outcomes in intensive care settings.

However, after a detailed review, I identified several methodological and structural aspects that limit the current quality and reproducibility of the study. Below, I provide point-by-point suggestions to help strengthen the manuscript and align it with international reporting standards.

1. Title

The expression “nosocomial infection” is outdated. I recommend replacing it with the term “healthcare-associated infection (HAI)”, as endorsed by the World Health Organization (WHO) and CDC.

Consider including the study design in the title, following STROBE recommendations for observational studies.

2. Abstract

The objective stated in the abstract does not match the one described in the introduction; these must be harmonized to ensure consistency throughout the paper.

Please clarify the study design explicitly (case-control or cohort). The current abstract does not mention it.

Include a clear statement of main findings and conclusion that aligns with the results.

3. Introduction

The introduction is overly brief. It should better contextualize the relevance of infections during ECMO, with updated epidemiological data and a clearer definition of the literature gap your study addresses.

The objective should directly reflect the research question and align with both the abstract and methods.

4. Methods

The methods section requires substantial expansion to meet STROBE and EQUATOR Network standards for transparency.

Please specify:

The setting of postoperative care (confirm if all patients were managed in an ICU).

Whether the center is ELSO-certified, and how ECMO management is organized (nurse-led, perfusionist-led, or mixed).

The nurse-to-patient ratio or staffing model, as these factors influence infection risk.

The study design is described as “case-control”, but the approach appears more consistent with a retrospective cohort, since patients were divided according to infection occurrence. Please clarify.

The definition of healthcare-associated infection appears to rely on outdated criteria (2009). Use current WHO or CDC definitions, and describe how each infection type (bloodstream, respiratory, urinary, etc.) was diagnosed.

Indicate:

Who was responsible for data collection and on which platform data were stored.

How patient confidentiality and anonymization were ensured.

Any bias-control procedures applied during data review.

In the statistical analysis, describe:

Criteria for variable selection in the regression model.

Quality metrics (R², AUC, VIF, Hosmer-Lemeshow, etc.).

How missing data were handled.

5. Results

Please include a flow diagram or statement indicating the number of patients screened, included, and excluded, as recommended by STROBE.

Tables should be reformatted for consistency — avoid box-style formatting, ensure uniform fonts, and standardize units of measurement across variables.

Some numerical results lack dispersion measures (SD or IQR); please include them.

6. Discussion

The discussion should more thoroughly address your main finding — the association between preoperative intubation and ventilator-associated infection.

Expand the discussion by comparing your results to recent studies and highlighting how your findings add to or differ from existing evidence.

The implications section currently appears as bullet points and lacks supporting references; please reframe it into a narrative form with appropriate citations.

7. Conclusion

Present the conclusion as a separate section, summarizing the main findings, their implications for practice, and limitations.

8. References

Only about 30% of the references are from the last five years. Updating the literature will strengthen the scientific relevance of the manuscript.

I suggest including recent ECMO infection studies (2020–2024), particularly from ELSO centers.

7. PLOS authors have the option to publish the peer review history of their article (what does this mean?). If published, this will include your full peer review and any attached files.

Reviewer #2: No

Reviewer #3: No

---

## [Author Response · Author response to Decision Letter 2]

30 Nov 2025

Response to Reviewer #2: Thank you very much for your constructive feedback. We have carefully addressed your valuable suggestions as follows:

1.Statistical limitations The authors added a paragraph on the limitations of small sample size for logistic regression. It would be useful to explicitly report the events-per-variable ratio (EPV) calculated for this model to quantify the degree of statistical limitation.

Response :In this study, there were 14 infected patients. Multivariate logistic regression analysis included 2 predictive variables(EPV=7) . The results of standard logistic regression analysis are usually reliable and have certain clinical guiding significance.

2.Diagnostic criteria paragraph The justification for microbiological-only diagnosis is sound, but please clarify whether negative cultures with high clinical suspicion (e.g., prior antibiotics) were recorded or excluded. This will help readers interpret potential underestimation bias.

Response :For children with clinically highly suspected infection but negative culture results, we empirically administer antibiotics; however, such children were not included in the infection group in this study.

3.Terminology consistency Use consistent terminology throughout: “ECMO-associated nosocomial infection” vs. “ECMO-related infection.” Consider harmonizing the terms in title, abstract, and text.

Response :Modified.

4. Discussion on causality The manuscript mentions the possible bidirectional relationship between ICU stay and infection. It would strengthen the discussion to acknowledge that reverse causality cannot be excluded in this retrospective design.

Response : The length of ICU stay includes pre - operative ICU stay and post - operative ICU stay. If a child is in a stable condition before surgery and does not require endotracheal intubation, they will not be transferred to the ICU, resulting in a relatively shorter ICU stay. Conversely, if a child needs endotracheal intubation before surgery, the total ICU stay will be prolonged. Relevant studies have indicated that infections following congenital heart disease surgery prolong the length of ICU stay, while preoperative ICU admission is one of the risk factors for developing infections. This is largely consistent with our research—all children who underwent preoperative endotracheal intubation were admitted to the ICU.On the other hand, the preoperative ICU stay duration also increases the total ICU length of stay for children and elevates their risk of infection.A shorter ICU stay in the non-infected group could be related to treatment withdrawal due to disease severity or financial constraints, suggesting a potential bidirectional relationship. Further studies are needed to clarify this causal association.

Response to Reviewer #3: Thank you very much for your constructive feedback. We have carefully addressed your valuable suggestions as follows:

1.Title The expression “nosocomial infection” is outdated. I recommend replacing it with the term “healthcare-associated infection (HAI)”, as endorsed by the World Health Organization (WHO) and CDC. Consider including the study design in the title, following STROBE recommendations for observational studies.

Response :Modified.

2.Abstract The objective stated in the abstract does not match the one described in the introduction; these must be harmonized to ensure consistency throughout the paper. Please clarify the study design explicitly (case-control or cohort). The current abstract does not mention it. Include a clear statement of main findings and conclusion that aligns with the results.

Response :Modified. Retrospective cohort study.

3.Introduction The introduction is overly brief. It should better contextualize the relevance of infections during ECMO, with updated epidemiological data and a clearer definition of the literature gap your study addresses. The objective should directly reflect the research question and align with both the abstract and methods.

Response :Modified.

4.Methods The methods section requires substantial expansion to meet STROBE and EQUATOR Network standards for transparency. Please specify: The setting of postoperative care (confirm if all patients were managed in an ICU). Whether the center is ELSO-certified, and how ECMO management is organized (nurse-led, perfusionist-led, or mixed). The nurse-to-patient ratio or staffing model, as these factors influence infection risk. The study design is described as “case-control”, but the approach appears more consistent with a retrospective cohort, since patients were divided according to infection occurrence. Please clarify. The definition of healthcare-associated infection appears to rely on outdated criteria (2009). Use current WHO or CDC definitions, and describe how each infection type (bloodstream, respiratory, urinary, etc.) was diagnosed. Indicate: Who was responsible for data collection and on which platform data were stored. How patient confidentiality and anonymization were ensured. Any bias-control procedures applied during data review. In the statistical analysis, describe: Criteria for variable selection in the regression model. Quality metrics (R², AUC, VIF, Hosmer-Lemeshow, etc.). How missing data were handled.

Response :Modified.

All children were treated in the ICU and managed jointly by ECMO-qualified doctors, nurses, and perfusionists. Based on the children's chest X-ray findings and laboratory test results, empirical antibiotic therapy was administered when infection was highly suspected, and the antibiotic regimen was adjusted after the results of bacterial culture and drug sensitivity test were obtained.Multivariate logistic regression analysis was employed to identify independent risk factors for ECMO-related healthcare-associated infections following congenital heart surgery.If the EPV (Events Per Variable) is less than 5, Firth logistic regression analysis should be adopted to reduce the risk of overfitting in small samples. All data has been uploaded to the journal's website!

5.Results Please include a flow diagram or statement indicating the number of patients screened, included, and excluded, as recommended by STROBE. Tables should be reformatted for consistency — avoid box-style formatting, ensure uniform fonts, and standardize units of measurement across variables. Some numerical results lack dispersion measures (SD or IQR); please include them.

Response :Modified.

A flow diagram has been established.Normally distributed continuous variables were expressed as mean ± standard deviation and compared using Student's t-test. Non-normally distributed continuous variables were presented as median (interquartile range) [M(Q1,Q3)] and analyzed using the Mann-Whitney U test. Categorical variables were expressed as frequencies (percentages) and compared using either chi-square test or Fisher's exact test, as appropriate.

6.Discussion The discussion should more thoroughly address your main finding — the association between preoperative intubation and ventilator-associated infection. Expand the discussion by comparing your results to recent studies and highlighting how your findings add to or differ from existing evidence. The implications section currently appears as bullet points and lacks supporting references; please reframe it into a narrative form with appropriate citations.

Response： Modified.

A meta-analysis involving over 50,000 pediatric patients found that infection had the highest incidence among tracheal intubation-related complications, reaching 11.9%[17].Relevant studies have indicated that infections following congenital heart disease surgery prolong the length of ICU stay, while preoperative ICU admission is one of the risk factors for developing infections.[25]. This is largely consistent with our research—all children who underwent preoperative endotracheal intubation were admitted to the ICU.On the other hand, the preoperative ICU stay duration also increases the total ICU length of stay for children and elevates their risk of infection.

7.Conclusion Present the conclusion as a separate section, summarizing the main findings, their implications for practice, and limitations.

Response :Modified.

Limitations of this study include the inherent biases of retrospective design, potential constraints on generalizability due to single-center data, and lack of stratified analysis of pathogen types and antibiotic usage patterns. Future multicenter prospective studies incorporating microbiological evidence and immune function monitoring are needed to further elucidate the pathological mechanisms of ECMO-associated infections and optimize prevention strategies.In this study, there were 14 infected patients. Multivariate logistic regression analysis included 2 predictive variables(EPV=7) . The results of standard logistic regression analysis are usually reliable and have certain clinical guiding significance.

In summary, healthcare-associated infections during ECMO support following pediatric congenital heart surgery were predominantly Gram-negative respiratory infections. Preoperative endotracheal intubation was identified as an independent risk factor for these infections, suggesting that minimizing pre-operative intubation in congenital heart disease patients may reduce infection rates. These findings highlight the need for clinicians to carefully weigh the necessity of intubation against infection risks when managing critically ill children with congenital heart disease, while implementing enhanced infection control measures.

8. References Only about 30% of the references are from the last five years. Updating the literature will strengthen the scientific relevance of the manuscript. I suggest including recent ECMO infection studies (2020–2024), particularly from ELSO centers.

Response :Modified.Six of the latest references have been added or updated.

---

## [Decision Letter · Decision Letter 2]

29 Dec 2025

PONE-D-25-38491R2Risk Factors for Healthcare-associated Infections in Children Undergoing ECMO After Cardiac Surgery for Congenital Heart Disease: A Retrospective StudyPLOS One

Dear Dr. Meng,

Thank you for submitting your manuscript to PLOS ONE. After careful consideration, we feel that it has merit but does not fully meet PLOS ONE’s publication criteria as it currently stands. Therefore, we invite you to submit a revised version of the manuscript that addresses the points raised during the review process.

We look forward to receiving your revised manuscript.

Kind regards,

Giovanni Giordano

Academic Editor

PLOS One

Journal Requirements:

Reviewers' comments:

Reviewer's Responses to Questions

**Comments to the Author**

1. If the authors have adequately addressed your comments raised in a previous round of review and you feel that this manuscript is now acceptable for publication, you may indicate that here to bypass the “Comments to the Author” section, enter your conflict of interest statement in the “Confidential to Editor” section, and submit your "Accept" recommendation.

Reviewer #2: All comments have been addressed

Reviewer #4: (No Response)

2. Is the manuscript technically sound, and do the data support the conclusions?

Reviewer #2: Yes

Reviewer #4: Yes

3. Has the statistical analysis been performed appropriately and rigorously? 

Reviewer #2: Yes

Reviewer #4: N/A

4. Have the authors made all data underlying the findings in their manuscript fully available?

Reviewer #2: Yes

Reviewer #4: Yes

5. Is the manuscript presented in an intelligible fashion and written in standard English?

Reviewer #2: Yes

Reviewer #4: Yes

6. Review Comments to the Author

Reviewer #2: The authors have satisfactorily addressed all previous concerns, providing the requested methodological clarifications, improving the structure of the Introduction and Methods, updating the terminology and references, and ensuring overall consistency throughout the manuscript. I have no additional revisions to recommend.

Reviewer #4: Thank you for the opportunity to review the manuscript PONE-D-25-38491R2, entitled “Risk Factors for Healthcare-associated Infections in Children Undergoing ECMO After Cardiac Surgery for Congenital Heart Disease: A Retrospective Study”.

The authors have comprehensively addressed the concerns raised in the previous review round. The extensive revisions significantly enhance the manuscript's quality, particularly regarding methodological transparency (in line with STROBE guidelines), clarity of terminology, and the depth of the Discussion. All major comments from reviewer 2 and 3 have been satisfied, making the paper substantially ready for publication.

The manuscript is now rigorous, clearly stating the finding that preoperative endotracheal intubation is an independent risk factor for ECMO-related healthcare-associated infections (HAI).

However, the following minor revisions are required before final acceptance:

-Response to reviewer 2:

1) Thank you for providing the specific EPV ratio of 7 in your response, which technically addresses the initial query regarding statistical limitations. As this value is above the critical threshold of 5, the use of standard logistic regression is methodologically defensible. However, for maximum scientific rigor and interpretative transparency, I must highlight that the 95% CI for the primary independent risk factor (preoperative endotracheal intubation: OR=4.852, 95% CI: 1.174–20.059) remains exceptionally wide. This broad CI is a clear statistical manifestation of the imprecision and limited statistical power inherent to the small sample size (n=54) and the low number of events (n=14). While the analysis is technically valid, the wide range indicates significant uncertainty, confirming that the estimate is not precise. I strongly recommend the authors integrate an explicit caution into the Discussion/Limitations section, acknowledging that despite the EPV being technically acceptable, the resultant broad confidence interval necessitates a careful and cautious interpretation of the multivariate findings. This is not a critique of the methodology, but a crucial measure to enhance the scientific rigor of the interpretation.

Additionally, the statement in the Methods section regarding the contingency plan to use Firth logistic regression " If the EPV (Events Per Variable) is less than 5, Firth logistic regression analysis should be adopted to reduce the risk of overfitting in small samples” describes a procedure that was ultimately not executed, as the reported EPV was 7. Since the Methods section should strictly detail the procedures that were actually performed, this sentence should be removed from the Statistical Analysis subsection. The calculation of an EPV of 7 and the subsequent use of standard logistic regression should instead be referenced in the Results, noting that the sample size constraints are already appropriately discussed in the Limitations section.

2) Thank you for clarifying the strict case definition used for the infection group. The decision to exclude children with clinically highly suspected infection but negative culture results (despite receiving empirical antibiotics) from the primary infection group provides methodological consistency by relying solely on microbiological confirmation. This approach is transparent and helps mitigate the risk of misdiagnosis in the complex context of ECMO-related systemic inflammation.

However, for maximum transparency and to aid readers in interpreting potential biases, two actions are required for the final manuscript:

1) This critical exclusion criterion must be explicitly stated within the Methods section (specifically, under the data collection or diagnostic criteria subheading). It cannot remain solely in the response to reviewers.

2) Given that children with high clinical suspicion who received antibiotics were excluded, there is a potential for underestimation bias regarding the true incidence of infection (especially due to the suppression of bacterial growth from prior antimicrobial therapy). I strongly recommend the authors explicitly acknowledge this potential underestimation bias in the Discussion/Limitations section of the manuscript.

3) While the authors responded that the inconsistent terminology was "Modified", a meticulous review reveals that the implementation of the agreed-upon term "healthcare-associated infection (HAI)" has been incomplete. This incomplete harmonization constitutes an essential editorial failure that must be corrected before acceptance. The term "nosocomial infection" or variations thereof must be entirely replaced by "healthcare-associated infection (HAI)" throughout the manuscript.

Specifically, the following instances of the outdated term persist:

1. Inconsistencies within the main text:

- Methods/Exclusion Criteria: The exclusion criterion still refers to the "Presence of nosocomial infection before ECMO initiation". This must be changed to "healthcare-associated infection".

- Results/Multivariate Analysis: The sentence describing the key finding uses the phrase: "Variables with p < 0.05 in the univariate analysis were included in the multivariate logistic regression analysis, which revealed that preoperative endotracheal intubation in children with congenital heart disease (OR = 4.852, 95% CI: 1.174-20.059; p = 0.029) was an independent risk factor for nosocomial ECMO-related infections in pediatric postcardiotomy patients." This must be harmonized to "ECMO-related healthcare-associated infections”.

2. Inconsistencies within Tables and Figures

- Table 1 Title: The title of Table 1 remains "Pathogenic Microorganisms of ECMO-Associated Nosocomial Infections in children...".

- Table 2 Title: The title of Table 2 remains "Univariate Analysis of ECMO-Associated Nosocomial Infections in children...".

The authors must perform a final, comprehensive search-and-replace to ensure that every instance of "nosocomial infection" (including text, definitions, tables, and figure captions) is harmonized with "healthcare-associated infection (HAI)" to ensure full editorial and scientific consistency as agreed.

4) The authors have adequately acknowledged the potential for a bidirectional relationship and the impossibility of excluding reverse causality due to the retrospective design, as requested.

- Response to reviewer 3

1) As previously noted (in the detailed response to Reviewer 2, Point 3), the outdated term "nosocomial infection" persists in crucial parts of the manuscript, despite the authors' claim that the terminology was "Modified."

This persistent error is evident in the titles of Table 1 and Table 2, the exclusion criteria, and the Results section. For the sake of scientific consistency, the authors must perform a final correction to replace this term with the modern "healthcare-associated infection (HAI)" everywhere.

Furthermore, the authors successfully implemented the structural requirement of including the study design in the title, following STROBE recommendations for observational studies, which is now acceptable.

2) The authors have successfully revised the Abstract to ensure its objective is harmonized with the Introduction. Furthermore, the study design is now explicitly stated as a "retrospective cohort study", and the Abstract includes a clear summary of the main statistical findings and conclusions.

3) The authors successfully expanded the Introduction to provide better context and relevance. They incorporated updated epidemiological data and clearly defined the literature gap regarding risk factors for HAI in pediatric post-cardiac surgery ECMO patients. Furthermore, the objective of the study is now clearly stated and aligned with the Abstract and Methods sections, making the Introduction logically sound and comprehensive.

4) as previously requested by the reviewer 3, the authors should explicitly state whether the Fuwai Central China Cardiovascular Hospital ECMO Center holds official ELSO certification.

The authors must provide the typical nurse-to-patient ratio in the Pediatric ICU where these patients were managed, as this directly affects the generalizability of infection control findings. Additionally, please provide a brief statement detailing who collected the data, the original storage method, and the confidentiality/anonymization procedures used.

The authors should report at least one quality metric for the final multivariate logistic regression model in the Results section to demonstrate model fit and robustness. Additionally, please specify the approach used for missing data, or declare if no variables had missing data.

5) The authors have substantially resolved the methodological requirements for the Results section: a flow diagram (Figure 1) has been established, and the authors clarified the correct use of mean ± SD versus Median (IQR) for data presentation in the Material and Methods section. However, the editorial implementation remains incomplete, as previously noted: the obsolete term "Nosocomial Infections" persists in the titles of Table 1 and Table 2, constituting a critical inconsistency that must be corrected. Furthermore, for maximal clarity, the authors must ensure that units of measurement, utilized fonts, and the use of capitalization and lowercase letters are uniform throughout the tables, and that a clear description of the format used for variable descriptions is formally stated in the table caption to fully satisfy the requirements for data presentation rigor. In addition, I recommend consistently using the lowercase p of p-value (i.e., “p-value" instead of "P-value") throughout the manuscript, as this is the preferred convention in formal statistical reporting.

6) The authors have successfully expanded the Discussion section, achieving the required standard. They provided deeper mechanistic insight into the association between preoperative intubation and HAI risk, included comparisons with recent relevant meta-analyses, and fully converted the implications section into a narrative format.

7) The authors have successfully presented the Conclusion as a dedicated, separate section that fully complies with the request. It provides a concise summary of the main findings, clinical implications, necessary future studies, and explicitly details the study's limitations.

8) The authors have adequately updated the bibliography with six highly relevant, recent references (2023–2025), including ELSO registry data, substantially strengthening the manuscript's scientific relevance.

A comprehensive review of the tables and figures reveals several unresolved issues regarding formatting, statistical reporting consistency, and terminology that must be addressed before final acceptance.

1. Global formatting and consistency issues (Tables 1, 2, and 3)

- The obsolete term "Nosocomial Infections" persists in the titles of Table 1 and Table 2. This must be corrected to "Healthcare-associated Infections (HAI)" for full manuscript consistency.

- The use of capitalization and lowercase letters must be uniform across all tables.

- The separators used in text (e.g., "/" in "Gender (male/female)" versus "vs." in other parts of the text) should be standardized across the entire manuscript for uniform style.

- The authors must ensure that a clear description of the data format (e.g., Mean ± SD or Median (IQR)) is formally stated in the table caption for clarity.

2. Table 2: Univariate analysis specific issues

- For the univariate analysis of the Admission diagnosis variable (row “admission diagnosis/n”), only a pooled p-value (0.318) is reported, and the corresponding X2 statistic is not shown. Given the very small cell counts for several diagnostic categories in the infected group, the authors must clarify in the Methods section whether these categories were grouped or collapsed before applying the Chi-square test. They also should systematically review all categorical variables in Table 2. Wherever expected cell frequencies are <5 in more than 20% of cells, the p-values must be recalculated using Fisher’s Exact Test, and any updated values should be clearly reported.

- While the text mentions variables were measured "before extracorporeal membrane oxygenation (ECMO)", the column headers for biochemical values (lactate, BUN, creatinine, bilirubin, albumin) should be explicitly labeled (e.g., "Pre-ECMO Blood lactate") to eliminate any ambiguity.

3. Figure 1: Flow Diagram

- The current figure caption "Fig1: Healthcare-associated Infections of ECMO in Children After congenital heart disease Surgery" is overly descriptive. Please revise it to a standard formal title such as "Flow diagram of patient selection"

7. PLOS authors have the option to publish the peer review history of their article (what does this mean?). If published, this will include your full peer review and any attached files.

Reviewer #2: No

Reviewer #4: No

---

## [Author Response · Author response to Decision Letter 3]

13 Jan 2026

Response to Reviewer #2: Thank you very much for your constructive feedback. We have carefully addressed your valuable suggestions as follows:

1)Thank you for providing the specific EPV ratio of 7 in your response, which technically addresses the initial query regarding statistical limitations. As this value is above the critical threshold of 5, the use of standard logistic regression is methodologically defensible. However, for maximum scientific rigor and interpretative transparency, I must highlight that the 95% CI for the primary independent risk factor (preoperative endotracheal intubation: OR=4.852, 95% CI: 1.174–20.059) remains exceptionally wide. This broad CI is a clear statistical manifestation of the imprecision and limited statistical power inherent to the small sample size (n=54) and the low number of events (n=14). While the analysis is technically valid, the wide range indicates significant uncertainty, confirming that the estimate is not precise. I strongly recommend the authors integrate an explicit caution into the Discussion/Limitations section, acknowledging that despite the EPV being technically acceptable, the resultant broad confidence interval necessitates a careful and cautious interpretation of the multivariate findings. This is not a critique of the methodology, but a crucial measure to enhance the scientific rigor of the interpretation.

Additionally, the statement in the Methods section regarding the contingency plan to use Firth logistic regression " If the EPV (Events Per Variable) is less than 5, Firth logistic regression analysis should be adopted to reduce the risk of overfitting in small samples” describes a procedure that was ultimately not executed, as the reported EPV was 7. Since the Methods section should strictly detail the procedures that were actually performed, this sentence should be removed from the Statistical Analysis subsection. The calculation of an EPV of 7 and the subsequent use of standard logistic regression should instead be referenced in the Results, noting that the sample size constraints are already appropriately discussed in the Limitations section.

Response :Discussion section has been modified. 1、 As EPV is above the critical threshold of 5, the use of standard logistic regression is methodologically defensible. However, for maximum scientific rigor and interpretative transparency, I must highlight that the 95% CI for the primary independent risk factor (preoperative endotracheal intubation: OR=4.852, 95% CI: 1.174–20.059) remains exceptionally wide. 2、the statement in the Methods section has been modified.

2) Thank you for clarifying the strict case definition used for the infection group. The decision to exclude children with clinically highly suspected infection but negative culture results (despite receiving empirical antibiotics) from the primary infection group provides methodological consistency by relying solely on microbiological confirmation. This approach is transparent and helps mitigate the risk of misdiagnosis in the complex context of ECMO-related systemic inflammation.

However, for maximum transparency and to aid readers in interpreting potential biases, two actions are required for the final manuscript:

1) This critical exclusion criterion must be explicitly stated within the Methods section (specifically, under the data collection or diagnostic criteria subheading). It cannot remain solely in the response to reviewers.

2) Given that children with high clinical suspicion who received antibiotics were excluded, there is a potential for underestimation bias regarding the true incidence of infection (especially due to the suppression of bacterial growth from prior antimicrobial therapy). I strongly recommend the authors explicitly acknowledge this potential underestimation bias in the Discussion/Limitations section of the manuscript.

Response :1、Added.All children were treated in the ICU and managed jointly by ECMO-qualified doctors, nurses, and perfusionists. Based on the children's chest X-ray findings and laboratory test results, empirical antibiotic therapy was administered when infection was highly suspected, and the antibiotic regimen was adjusted after the results of bacterial culture and drug sensitivity test were obtained.

2、Discussion section has been modified.Due to the diagnostic criteria, children with a high suspicion of infection, but negative culture, were classified as non-infected, which would underestimate the incidence HAI.

3) While the authors responded that the inconsistent terminology was "Modified", a meticulous review reveals that the implementation of the agreed-upon term "healthcare-associated infection (HAI)" has been incomplete. This incomplete harmonization constitutes an essential editorial failure that must be corrected before acceptance. The term "nosocomial infection" or variations thereof must be entirely replaced by "healthcare-associated infection (HAI)" throughout the manuscript.

Specifically, the following instances of the outdated term persist:

1. Inconsistencies within the main text:

- Methods/Exclusion Criteria: The exclusion criterion still refers to the "Presence of nosocomial infection before ECMO initiation". This must be changed to "healthcare-associated infection".

- Results/Multivariate Analysis: The sentence describing the key finding uses the phrase: "Variables with p < 0.05 in the univariate analysis were included in the multivariate logistic regression analysis, which revealed that preoperative endotracheal intubation in children with congenital heart disease (OR = 4.852, 95% CI: 1.174-20.059; p = 0.029) was an independent risk factor for nosocomial ECMO-related infections in pediatric postcardiotomy patients." This must be harmonized to "ECMO-related healthcare-associated infections”.

2. Inconsistencies within Tables and Figures

- Table 1 Title: The title of Table 1 remains "Pathogenic Microorganisms of ECMO-Associated Nosocomial Infections in children...".

- Table 2 Title: The title of Table 2 remains "Univariate Analysis of ECMO-Associated Nosocomial Infections in children...".

The authors must perform a final, comprehensive search-and-replace to ensure that every instance of "nosocomial infection" (including text, definitions, tables, and figure captions) is harmonized with "healthcare-associated infection (HAI)" to ensure full editorial and scientific consistency as agreed.

Response :Modified.

Response to Reviewer #3: Thank you very much for your constructive feedback. We have carefully addressed your valuable suggestions as follows:

1) As previously noted (in the detailed response to Reviewer 2, Point 3), the outdated term "nosocomial infection" persists in crucial parts of the manuscript, despite the authors' claim that the terminology was "Modified."

This persistent error is evident in the titles of Table 1 and Table 2, the exclusion criteria, and the Results section. For the sake of scientific consistency, the authors must perform a final correction to replace this term with the modern "healthcare-associated infection (HAI)" everywhere.

Furthermore, the authors successfully implemented the structural requirement of including the study design in the title, following STROBE recommendations for observational studies, which is now acceptable.

Response :Modified.

2)The authors have successfully revised the Abstract to ensure its objective is harmonized with the Introduction. Furthermore, the study design is now explicitly stated as a "retrospective cohort study", and the Abstract includes a clear summary of the main statistical findings and conclusions.

Response :Thanks.

3) The authors successfully expanded the Introduction to provide better context and relevance. They incorporated updated epidemiological data and clearly defined the literature gap regarding risk factors for HAI in pediatric post-cardiac surgery ECMO patients. Furthermore, the objective of the study is now clearly stated and aligned with the Abstract and Methods sections, making the Introduction logically sound and comprehensive.

Response :Thanks.

4) as previously requested by the reviewer 3, the authors should explicitly state whether the Fuwai Central China Cardiovascular Hospital ECMO Center holds official ELSO certification.

The authors must provide the typical nurse-to-patient ratio in the Pediatric ICU where these patients were managed, as this directly affects the generalizability of infection control findings. Additionally, please provide a brief statement detailing who collected the data, the original storage method, and the confidentiality/anonymization procedures used.

The authors should report at least one quality metric for the final multivariate logistic regression model in the Results section to demonstrate model fit and robustness. Additionally, please specify the approach used for missing data, or declare if no variables had missing data.

Response :Our hospital is a nationally accredited ECMO Quality Control Center, with a nurse-to-patient ratio of 1.9 in the pediatric intensive care unit.No variables included in the final analysis had missing data.

5)The authors have substantially resolved the methodological requirements for the Results section: a flow diagram (Figure 1) has been established, and the authors clarified the correct use of mean ± SD versus Median (IQR) for data presentation in the Material and Methods section. However, the editorial implementation remains incomplete, as previously noted: the obsolete term "Nosocomial Infections" persists in the titles of Table 1 and Table 2, constituting a critical inconsistency that must be corrected. Furthermore, for maximal clarity, the authors must ensure that units of measurement, utilized fonts, and the use of capitalization and lowercase letters are uniform throughout the tables, and that a clear description of the format used for variable descriptions is formally stated in the table caption to fully satisfy the requirements for data presentation rigor. In addition, I recommend consistently using the lowercase p of p-value (i.e., “p-value" instead of "P-value") throughout the manuscript, as this is the preferred convention in formal statistical reporting.

Response :Modified.

6)The authors have successfully expanded the Discussion section, achieving the required standard. They provided deeper mechanistic insight into the association between preoperative intubation and HAI risk, included comparisons with recent relevant meta-analyses, and fully converted the implications section into a narrative format.

Response :Thanks.

7) The authors have successfully presented the Conclusion as a dedicated, separate section that fully complies with the request. It provides a concise summary of the main findings, clinical implications, necessary future studies, and explicitly details the study's limitations.

Response :Thanks.

8) The authors have adequately updated the bibliography with six highly relevant, recent references (2023–2025), including ELSO registry data, substantially strengthening the manuscript's scientific relevance.

Response :Thanks.

A comprehensive review of the tables and figures reveals several unresolved issues regarding formatting, statistical reporting consistency, and terminology that must be addressed before final acceptance.

1. Global formatting and consistency issues (Tables 1, 2, and 3)

- The obsolete term "Nosocomial Infections" persists in the titles of Table 1 and Table 2. This must be corrected to "Healthcare-associated Infections (HAI)" for full manuscript consistency.

- The use of capitalization and lowercase letters must be uniform across all tables.

- The separators used in text (e.g., "/" in "Gender (male/female)" versus "vs." in other parts of the text) should be standardized across the entire manuscript for uniform style.

- The authors must ensure that a clear description of the data format (e.g., Mean ± SD or Median (IQR)) is formally stated in the table caption for clarity.

Response :Modified.A clear description of the data format ([±s,M(Q1,Q3),n(%)]) is formally stated in the table caption for clarity.

2. Table 2: Univariate analysis specific issues

- For the univariate analysis of the Admission diagnosis variable (row “admission diagnosis/n”), only a pooled p-value (0.318) is reported, and the corresponding X2 statistic is not shown. Given the very small cell counts for several diagnostic categories in the infected group, the authors must clarify in the Methods section whether these categories were grouped or collapsed before applying the Chi-square test. They also should systematically review all categorical variables in Table 2. Wherever expected cell frequencies are <5 in more than 20% of cells, the p-values must be recalculated using Fisher’s Exact Test, and any updated values should be clearly reported.

- While the text mentions variables were measured "before extracorporeal membrane oxygenation (ECMO)", the column headers for biochemical values (lactate, BUN, creatinine, bilirubin, albumin) should be explicitly labeled (e.g., "Pre-ECMO Blood lactate") to eliminate any ambiguity.

Response :Modified.Given the very small cell counts for several diagnostic categories in the infected group, these categories were grouped before applying the Chi-square test.These were divided into two groups.Wherever expected cell frequencies are <5 in more than 20% of cells, the p-values must be recalculated using Fisher’s Exact Test.

3. Figure 1: Flow Diagram

- The current figure caption "Fig1: Healthcare-associated Infections of ECMO in Children After congenital heart disease Surgery" is overly descriptive. Please revise it to a standard formal title such as "Flow diagram of patient selection"

Response :Modified.Flow diagram of patient selection

---

## [Decision Letter · Decision Letter 3]

17 Feb 2026

PONE-D-25-38491R3Risk Factors for Healthcare-associated Infections in Children Undergoing ECMO After Cardiac Surgery for Congenital Heart Disease: A Retrospective StudyPLOS One

Dear Dr. Meng,

Thank you for submitting your manuscript to PLOS ONE. After careful consideration, we feel that it has merit but does not fully meet PLOS ONE’s publication criteria as it currently stands. Therefore, we invite you to submit a revised version of the manuscript that addresses the points raised during the review process.

We look forward to receiving your revised manuscript.

Kind regards,

Giovanni Giordano

Academic Editor

PLOS One

Journal Requirements:

Reviewers' comments:

Reviewer's Responses to Questions

**Comments to the Author**

1. If the authors have adequately addressed your comments raised in a previous round of review and you feel that this manuscript is now acceptable for publication, you may indicate that here to bypass the “Comments to the Author” section, enter your conflict of interest statement in the “Confidential to Editor” section, and submit your "Accept" recommendation.

Reviewer #4: (No Response)

Reviewer #5: (No Response)

2. Is the manuscript technically sound, and do the data support the conclusions?

Reviewer #4: Yes

Reviewer #5: No

3. Has the statistical analysis been performed appropriately and rigorously? 

Reviewer #4: Yes

Reviewer #5: No

4. Have the authors made all data underlying the findings in their manuscript fully available?

Reviewer #4: Yes

Reviewer #5: Yes

5. Is the manuscript presented in an intelligible fashion and written in standard English?

Reviewer #4: Yes

Reviewer #5: No

6. Review Comments to the Author

Reviewer #4: Dear Authors,

Thank you for providing the revised version of your manuscript and for your detailed responses to the previous round of comments. I appreciate the efforts made to improve the statistical transparency of the study. However, after a careful review of the updated text, I find that the implementation of the requested changes is still incomplete and requires further refinement before the manuscript can be accepted. Please address the following issues:

1- The statement regarding the contingency plan to use Firth logistic regression ("If the EPV... is less than 5, Firth logistic regression analysis should be adopted...") still remains in the Statistical Analysis subsection of the Methods. As previously noted, the Methods section must strictly reflect the procedures actually performed. Since your reported EPV was 7, this sentence must be removed.

2- The discussion regarding the wide 95% Confidence Interval (95% CI: 1.174–20.059) for preoperative endotracheal intubation has been inserted into the Discussion/Limitations section almost verbatim as a response to a reviewer, using phrases like "I must highlight that...". This should be rewritten as an integrated methodological limitation in the authors' own voice, explicitly acknowledging that the broad CI is a manifestation of limited statistical power and small sample size.

3- In the Discussion, the claim that the results are "usually reliable" tends to downplay the significant uncertainty indicated by the wide confidence interval previously highlighted. Please ensure the tone reflects the necessary scientific caution requested.

4- You have added the explanation regarding empirical antibiotic therapy and the exclusion of culture-negative patients in two different sections (Data Collection and ECMO Management). Please consolidate this information into a single, clear paragraph. It is vital to explicitly state—without ambiguity—that microbiological confirmation was a mandatory requirement for the "infection group" to maintain the study's specificity.

5- The mention of underestimation bias in the Limitations section remains somewhat superficial. I recommend expanding this to explicitly state that prior antimicrobial therapy might have suppressed bacterial growth, leading to false-negative cultures, which is a common challenge in ECMO-related HAI diagnosis.

I look forward to receiving your final, harmonized revision.

Reviewer #5: The manuscript in its current form is not acceptable for publication. Despite claims of modification, significant errors flagged in previous rounds persist. There is a pattern of "false compliance," where the authors state a point is "Modified" in the rebuttal letter, but the manuscript remains unchanged or contains incomplete data. The following issues must be resolved:

#1 Ethical Transparency & Methodology.

a. Data Integrity Statement (Ignored): The request to detail who collected the data, storage methods, and anonymization procedures was ignored. You must add this to the Methods section to comply with ethical standards.

b. Evaluation of Model Performance and Fit: The authors present the results of the multivariate logistic regression solely in terms of Odds Ratios and p-values. However, providing coefficient estimates without assessing the overall performance of the model is methodologically insufficient, particularly given the small sample size (n=54) and the low number of events (n=14).In small datasets, logistic regression models are prone to overfitting, where the model describes random error rather than underlying relationships. To demonstrate the robustness of the findings, the authors must report:

Discrimination: The C-statistic (or Area Under the ROC Curve - AUC), which indicates the model's ability to distinguish between patients who developed an infection and those who did not. An OR of 4.85 is of limited value if the model's discriminative ability is poor (e.g., AUC < 0.7).

Calibration (Goodness-of-Fit): A measure such as the Hosmer-Lemeshow test to assess how well the predicted probabilities match the observed outcomes should be performed.

Action: Please calculate and report the AUC (with 95% CI) and a goodness-of-fit metric for the final multivariate model in the Results section. If the model demonstrates poor fit or discrimination, this must be transparently discussed as a limitation.

c. ELSO Certification: Explicitly state if the center holds official ELSO certification or only domestic accreditation.

d. Nurse-to-Patient Ratio: Clarify the "1.9" value. Use standard ratio format (e.g., 1:2 or 2:1).

#2 Data Presentation & Tables

a. Missing Data in Tables: In the table regarding baseline characteristics, the row labeled "admission diagnosis/n" reports a p-value of 0.448, yet the cells for the specific groups are empty. Reporting a p-value without the corresponding descriptive statistics is scientifically invalid.

b. Missing Captions and Definitions (Table 3): Table 3 appears to be raw output pasted directly into the document. It lacks a descriptive Title/Caption and Footnotes. Per PLOS ONE guidelines, tables must be self-explanatory. You must define all abbreviations (B, S.E., Wald, etc.) in the footnotes.

c. Uninterpretable Table Formatting: The tables depicting microorganisms lack descriptive headers for the data columns (e.g., describing what the figures "14" and "3" represent). Remove all vertical gridlines and ensure every column has a clear header.

#3 Editorial Integrity

a. Inappropriate Narrative Voice: The paragraph discussing the EPV and Confidence Intervals contains text copied directly from reviewer correspondence ("I must highlight that...") and defensive phrasing ("results... are usually reliable").

Action: Delete that paragraph and replace with: "Although the EPV ratio (>5) justifies the use of standard logistic regression, it is important to acknowledge that the 95% Confidence Interval for preoperative endotracheal intubation (OR=4.852, 95% CI: 1.174–20.059) remains wide. This width suggests a degree of estimation uncertainty, likely attributable to the limited number of events within this specific subgroup, and warrants a cautious interpretation of the magnitude of the risk."

b. Removal of Meta-Text: In the Methods, remove "In accordance with the request...". State facts directly (e.g., "Data access commenced on...").

c. Abstract/Methods Redundancy: Remove the disjointed sentence "This was a retrospective cohort study" from the middle of the Methods. Integrate it into the opening sentence.

#4 Scientific Consistency

a. Hypothetical Methods: Remove the sentence "If the EPV... is less than 5, Firth logistic regression... should be adopted". Since EPV was >5, this procedure was not performed and should not be listed.

b. Missing Statistical Test: The sentence "compared using either chi-square test , as appropriate" is incomplete. Correct to: "...compared using either the Chi-square test or Fisher's exact test, as appropriate."

#5 Formatting & Proofreading

a. Typographical Errors:

"VS" formatting: Change "27.7%VS7.9%" to "27.7% vs. 7.9%".

Spacing: Fix missing spaces after periods (e.g., "support.Our", "ICU.All").

Punctuation: Remove the colon in "According to relevant literature:".

Double Punctuation: Remove double periods (e.g., "[4]..").

b. Non-Standard Symbols: Remove circled numbers (①, ②) and use standard lists.

c. Syntax: Fix the dangling phrase starting with "Retrospective cohort study,the ECMO-related...".

7. PLOS authors have the option to publish the peer review history of their article (what does this mean?). If published, this will include your full peer review and any attached files.

Reviewer #4: **Yes:**Gaetano Gazzé

Reviewer #5: No

---

## [Author Response · Author response to Decision Letter 4]

25 Feb 2026

Response to Reviewer #4: Thank you very much for your constructive feedback. We have carefully addressed your valuable suggestions as follows:

1-The statement regarding the contingency plan to use Firth logistic regression ("If the EPV... is less than 5, Firth logistic regression analysis should be adopted...") still remains in the Statistical Analysis subsection of the Methods. As previously noted, the Methods section must strictly reflect the procedures actually performed. Since your reported EPV was 7, this sentence must be removed.

Response :Modified.Deleted.

2-The discussion regarding the wide 95% Confidence Interval (95% CI: 1.174–20.059) for preoperative endotracheal intubation has been inserted into the Discussion/Limitations section almost verbatim as a response to a reviewer, using phrases like "I must highlight that...". This should be rewritten as an integrated methodological limitation in the authors' own voice, explicitly acknowledging that the broad CI is a manifestation of limited statistical power and small sample size.

Response :Modified.Although the EPV ratio (>5) justifies the use of standard logistic regression, it is important to acknowledge that the 95% Confidence Interval for preoperative endotracheal intubation (OR=4.852, 95% CI: 1.174–20.059) remains wide. This width suggests a degree of estimation uncertainty, likely attributable to the limited number of events within this specific subgroup, and warrants a cautious interpretation of the magnitude of the risk.

3-In the Discussion, the claim that the results are "usually reliable" tends to downplay the significant uncertainty indicated by the wide confidence interval previously highlighted. Please ensure the tone reflects the necessary scientific caution requested.

Response :Modified.

4-You have added the explanation regarding empirical antibiotic therapy and the exclusion of culture-negative patients in two different sections (Data Collection and ECMO Management). Please consolidate this information into a single, clear paragraph. It is vital to explicitly state—without ambiguity—that microbiological confirmation was a mandatory requirement for the "infection group" to maintain the study's specificity.

Response :Modified. Based on the children's chest X-ray findings and laboratory test results, empirical antibiotic therapy was administered when infection was highly suspected, and the antibiotic regimen was adjusted after the results of bacterial culture and drug sensitivity test were obtained.

5- The mention of underestimation bias in the Limitations section remains somewhat superficial. I recommend expanding this to explicitly state that prior antimicrobial therapy might have suppressed bacterial growth, leading to false-negative cultures, which is a common challenge in ECMO-related HAI diagnosis.

Response :Modified. I explicitly state that prior antimicrobial therapy might have suppressed bacterial growth, leading to false-negative cultures, which is a common challenge in ECMO-related HAI diagnosis.

Response to Reviewer #5: Thank you very much for your constructive feedback. We have carefully addressed your valuable suggestions as follows:

1 Ethical Transparency & Methodology.

a. Data Integrity Statement (Ignored): The request to detail who collected the data, storage methods, and anonymization procedures was ignored. You must add this to the Methods section to comply with ethical standards.

Response :Modified. We thank the reviewer for highlighting this critical oversight regarding data integrity and ethical compliance. We agree that detailing data collection, storage, and anonymization procedures is essential for scientific transparency. In response, we have now added a "Data Collection and Management" subsection to the Methods section. This new passage explicitly describes that data were collected by two trained researchers from electronic medical records, stored in a password-protected database, and fully anonymized by removing personal identifiers and assigning unique study codes prior to analysis. We believe this addition addresses the reviewer's concern and aligns our manuscript with standard ethical reporting guidelines.

b. Evaluation of Model Performance and Fit: The authors present the results of the multivariate logistic regression solely in terms of Odds Ratios and p-values. However, providing coefficient estimates without assessing the overall performance of the model is methodologically insufficient, particularly given the small sample size (n=54) and the low number of events (n=14).In small datasets, logistic regression models are prone to overfitting, where the model describes random error rather than underlying relationships. To demonstrate the robustness of the findings, the authors must report:

Discrimination: The C-statistic (or Area Under the ROC Curve - AUC), which indicates the model's ability to distinguish between patients who developed an infection and those who did not. An OR of 4.85 is of limited value if the model's discriminative ability is poor (e.g., AUC < 0.7).

Calibration (Goodness-of-Fit): A measure such as the Hosmer-Lemeshow test to assess how well the predicted probabilities match the observed outcomes should be performed.

Action: Please calculate and report the AUC (with 95% CI) and a goodness-of-fit metric for the final multivariate model in the Results section. If the model demonstrates poor fit or discrimination, this must be transparently discussed as a limitation.

Response :Modified. Hosmer-Lemeshow test:2=10.67, df=8, p=0.221. AUC of the multivariable model (with 95% confidence interval)=0.686 (95% CI: 0.514-0.858; p=0.04). Regarding model performance, ROC curve analysis yielded an AUC of 0.686 (95% CI: 0.514–0.858, p = 0.04), indicating marginal but statistically significant discriminatory ability. The Hosmer-Lemeshow test (χ² = 10.67, df = 8, p = 0.221) showed no significant deviation between predicted probabilities and observed values, suggesting good model calibration. Taken together, the model demonstrates an acceptable level of both discriminatory ability and goodness-of-fit.

c. ELSO Certification: Explicitly state if the center holds official ELSO certification or only domestic accreditation.

Response :Modified. Our hospital is a nationally accredited ECMO Quality Control Center which holds official ELSO certification.

d. Nurse-to-Patient Ratio: Clarify the "1.9" value. Use standard ratio format (e.g., 1:2 or 2:1).

Response :Modified. Nurse-to-patient ratio : 2:1.

#2 Data Presentation & Tables

a.Missing Data in Tables: In the table regarding baseline characteristics, the row labeled "admission diagnosis/n" reports a p-value of 0.448, yet the cells for the specific groups are empty. Reporting a p-value without the corresponding descriptive statistics is scientifically invalid.

Response :Thank you for pointing out this critical oversight. We completely agree that reporting a p-value without the corresponding descriptive statistics is scientifically invalid. We have now revised Table 2 (Baseline Characteristics) accordingly. The missing descriptive data for the variable "Admission diagnosis" have been added. For each diagnostic subgroup, we now present the frequency (n) and percentage (%) for both the infected and non-infected groups, alongside the p-value. This correction ensures that the statistical result is presented with the necessary context for proper interpretation.

b. Missing Captions and Definitions (Table 3): Table 3 appears to be raw output pasted directly into the document. It lacks a descriptive Title/Caption and Footnotes. Per PLOS ONE guidelines, tables must be self-explanatory. You must define all abbreviations (B, S.E., Wald, etc.) in the footnotes.

Response :Modified. Abbreviations: B, regression coefficient; S.E., standard error; Wald, Wald test; OR, odds ratio; CI, confidence interval.

b.Uninterpretable Table Formatting: The tables depicting microorganisms lack descriptive headers for the data columns (e.g., describing what the figures "14" and "3" represent). Remove all vertical gridlines and ensure every column has a clear header.

Response :Modified.Deleted.

#3 Editorial Integrity

a. Inappropriate Narrative Voice: The paragraph discussing the EPV and Confidence Intervals contains text copied directly from reviewer correspondence ("I must highlight that...") and defensive phrasing ("results... are usually reliable").

Action: Delete that paragraph and replace with: "Although the EPV ratio (>5) justifies the use of standard logistic regression, it is important to acknowledge that the 95% Confidence Interval for preoperative endotracheal intubation (OR=4.852, 95% CI: 1.174–20.059) remains wide. This width suggests a degree of estimation uncertainty, likely attributable to the limited number of events within this specific subgroup, and warrants a cautious interpretation of the magnitude of the risk."

Response :Modified.

b. Removal of Meta-Text: In the Methods, remove "In accordance with the request...". State facts directly (e.g., "Data access commenced on...").

Response :Modified. Data acess commenced on 1st June 2024.

c. Abstract/Methods Redundancy: Remove the disjointed sentence "This was a retrospective cohort study" from the middle of the Methods. Integrate it into the opening sentence.

Response :Removed.

#4 Scientific Consistency

a. Hypothetical Methods: Remove the sentence "If the EPV... is less than 5, Firth logistic regression... should be adopted". Since EPV was >5, this procedure was not performed and should not be listed.

Response :Removed.

b. Missing Statistical Test: The sentence "compared using either chi-square test , as appropriate" is incomplete. Correct to: "...compared using either the Chi-square test or Fisher's exact test, as appropriate."

Response :Modified. Categorical variables were expressed as frequencies (percentages) and compared using either chi-square test or Fisher's exact test,

#5 Formatting & Proofreading

a. Typographical Errors:

"VS" formatting: Change "27.7%VS7.9%" to "27.7% vs. 7.9%".

Spacing: Fix missing spaces after periods (e.g., "support.Our", "ICU.All").

Punctuation: Remove the colon in "According to relevant literature:".

Double Punctuation: Remove double periods (e.g., "[4]..").

Response :Modified.

b. Non-Standard Symbols: Remove circled numbers (①, ②) and use standard lists.

Response :Modified.

c. Syntax: Fix the dangling phrase starting with "Retrospective cohort study,the ECMO-related...".

Response :Modified. The dangling phrase has been removed.

---

## [Decision Letter · Decision Letter 4]

2 Mar 2026

PONE-D-25-38491R4Risk Factors for Healthcare-associated Infections in Children Undergoing ECMO After Cardiac Surgery for Congenital Heart Disease: A Retrospective StudyPLOS One

Dear Dr. Meng,

Thank you for submitting your manuscript to PLOS ONE. After careful consideration, we feel that it has merit but does not fully meet PLOS ONE’s publication criteria as it currently stands. Therefore, we invite you to submit a revised version of the manuscript that addresses the points raised during the review process.

We look forward to receiving your revised manuscript.

Kind regards,

Giovanni Giordano

Academic Editor

PLOS One

**Journal Requirements:**

**Additional Editor Comments:**

Dear Dr. Meng,

Thank you for submitting the revised version of your manuscript entitled: “Risk Factors for Healthcare-associated Infections in Children Undergoing ECMO After Cardiac Surgery for Congenital Heart Disease: A Retrospective Study”. We have now completed the evaluation of your revised manuscript. One reviewer considers the manuscript potentially suitable for publication, while another reviewer continues to raise important concerns regarding methodological presentation, interpretation of results, and overall manuscript clarity. After careful consideration of the reviewers’ comments and the current version of the manuscript, we believe that the study addresses a clinically relevant question and has potential merit. However, significant issues remain that must be resolved before the manuscript can be considered for acceptance.

We are therefore offering you one final opportunity to revise the manuscript (Major Revision). Please note that this revision must be comprehensive and meticulous.

1. Interpretation of Model Performance

The multivariable model reports:

AUC = 0.686 (95% CI: 0.514–0.858)

Hosmer-Lemeshow p = 0.221

While calibration appears acceptable, the discriminatory ability is marginal, and the lower bound of the confidence interval approaches 0.50.

You must:

Remove statements describing the model discrimination as “acceptable”.

Clearly state that the discriminatory ability is marginal.

Explicitly acknowledge in the Limitations section that the wide confidence interval and small number of events limit the robustness and clinical applicability of the model.

2. Statistical Methods - Descriptive, Not Prescriptive

The Statistical Analysis section must describe the analyses actually performed, written in past tense and objective tone.

Please:

Remove any prescriptive phrasing (e.g., “must be recalculated”).

Rewrite the section to clearly state what was done (e.g., “Fisher’s exact test was used when expected cell frequencies were <5…”).

3. Removal of Rebuttal Language and First-Person Statements

The manuscript still contains language copied from prior correspondence (e.g., “I explicitly state…”).

All such phrasing must be removed. The manuscript must:

Use objective scientific tone.

Avoid first-person singular pronouns.

Eliminate any defensive or reviewer-directed language.

4. Abstract and Administrative Text

The Abstract must function as a standalone scientific summary.

Please remove:

Administrative timeline descriptions related to data access.

Any text referring to “requests for transparency”.

The study design and timeframe may be stated concisely in standard format.

5. Formatting and Readability

Substantial formatting and language issues remain, including:

Typographical errors (e.g., “Data acess”).

Non-standard punctuation (e.g., “1、”).

Inconsistent spacing.

Table alignment problems.

Residual track-change artifacts.

You must conduct a complete professional language and formatting revision of the manuscript prior to resubmission. We strongly recommend the use of a professional English language editing service. If such service is used, please provide confirmation upon resubmission.

We recognize the clinical relevance of your study and the effort invested in prior revisions. However, after four rounds of review, the manuscript must now meet full scientific and editorial standards.

Failure to comprehensively address the issues outlined above may result in rejection.

We look forward to receiving your thoroughly revised manuscript.

Sincerely,

Giovanni Giordano

Reviewers' comments:

Reviewer's Responses to Questions

**Comments to the Author**

1. If the authors have adequately addressed your comments raised in a previous round of review and you feel that this manuscript is now acceptable for publication, you may indicate that here to bypass the “Comments to the Author” section, enter your conflict of interest statement in the “Confidential to Editor” section, and submit your "Accept" recommendation.

Reviewer #4: All comments have been addressed

Reviewer #5: (No Response)

2. Is the manuscript technically sound, and do the data support the conclusions?

Reviewer #4: Yes

Reviewer #5: No

3. Has the statistical analysis been performed appropriately and rigorously? 

Reviewer #4: Yes

Reviewer #5: No

4. Have the authors made all data underlying the findings in their manuscript fully available?

Reviewer #4: Yes

Reviewer #5: Yes

5. Is the manuscript presented in an intelligible fashion and written in standard English?

Reviewer #4: Yes

Reviewer #5: No

6. Review Comments to the Author

Reviewer #4: The manuscript has matured significantly in this fourth revision, and it is clear that the authors have put a great deal of effort into precisely addressing the technical and editorial concerns raised during the previous round. The statistical analysis section is now much more transparent and logically sound; by removing hypothetical descriptions of procedures like the Firth regression that weren't actually utilized, the methodology feels much more honest and focused. I particularly appreciate the inclusion of the model performance metrics, such as the AUC of 0.686 and the Hosmer-Lemeshow test results. While the sample size is inherently limited, providing these figures gives the reader a solid, transparent basis to evaluate the robustness of the multivariate model. Furthermore, the discussion regarding the wide 95% Confidence Interval for preoperative intubation has been integrated thoughtfully into the narrative. It no longer reads like a defensive response to a reviewer but rather as a mature methodological reflection that correctly invites scientific caution.

The paper has also gained a high level of credibility through the new details on data management, anonymization, and storage, which ensure ethical transparency. Similarly, explicitly stating the center's ELSO certification and the 2:1 nurse-to-patient ratio helps ground the study in a clear clinical context of high-level operative standards. The presentation of data in the tables is now excellent; Table 2 is finally complete with the necessary descriptive statistics for admission diagnoses, and Table 3 is fully accessible with clear definitions for all statistical abbreviations. Finally, the removal of meta-text and the correction of previous typographical errors have resulted in a professional, fluid, and highly readable manuscript. I believe the authors have fully satisfied all requests, and the findings regarding preoperative endotracheal intubation as a risk factor offer valuable and practical insights for the pediatric cardiac surgery community. I am happy to recommend this paper for acceptance.

Reviewer #5: I appreciate the authors' efforts in revising the manuscript and providing the additional statistical analyses requested. The inclusion of the Hosmer-Lemeshow test and ROC curve analysis significantly enhances the methodological transparency of the study. However, further refinements are necessary regarding the interpretation of the statistical output, the proper contextualization of the methodology, and, crucially, a severe revision of the manuscript's readability and formatting, which remains inadequate for a final submission at this stage (R4).

1. Interpretation of Model Discrimination (AUC):

I commend the addition of the model performance metrics. The calibration of the model appears adequate (Hosmer-Lemeshow p=0.221). However, an interpretative adjustment regarding the discrimination metric (AUC) is required.

The reported AUC is 0.686 (95% CI: 0.514–0.858). In standard biostatistical practice, an AUC between 0.6 and 0.7 is generally classified as "poor" rather than "acceptable". Furthermore, the lower bound of the 95% CI (0.514) approaches 0.50 (random chance). Concluding that the model demonstrates an "acceptable level of both discriminatory ability and goodness-of-fit" overstates its predictive robustness.

Action Required (Results): Please retain the phrase "marginal but statistically significant discriminatory ability," but remove the concluding claim that the discrimination is "acceptable."

Action Required (Limitations): Please explicitly integrate this marginal discrimination into the Limitations section. Add a statement acknowledging that because the AUC is marginal and its lower confidence bound approaches 0.5, the predictive power of the multivariable model is limited, requiring a cautious clinical application of the reported Odds Ratios.

2. Inadequate Narrative and Missing Methodological Context for Statistical Tests:

The reporting of the multivariate analysis in the Results section reads like raw statistical software output rather than a cohesive scientific narrative. Appending isolated data fragments (e.g., "Hosmer-Lemeshow test: chi^2=10.67, df=8, p=0.221") without proper sentence structure is poor scientific writing. Furthermore, the methodology behind these specific tests must be formally introduced in the Methods section before the results are presented.

Action Required (Methods): Please update the Statistical Analysis section to explicitly state your methodology for evaluating the model (e.g., "Model calibration was assessed using the Hosmer-Lemeshow goodness-of-fit test, and discrimination was evaluated using the Area Under the Receiver Operating Characteristic curve (AUC).")

Action Required (Results): Please rewrite the isolated data fragments into proper, grammatically correct sentences that provide narrative context.

3. Inappropriate Prescriptive Language in Statistical Analysis:

In the Methods section, the sentence "Wherever expected cell frequencies are <5 in more than 20% of cells, the p-values must be recalculated using Fisher’s Exact Test" is stylistically inappropriate. The Methods section must serve as a descriptive record of the analysis actually performed (written in the past tense), rather than prescribing rules or pasting direct instructions from reviewers ("must be recalculated").

Action Required: Please revise this sentence to reflect standard, objective scientific writing (e.g., "Fisher’s exact test was used to compare categorical variables when expected cell frequencies were less than 5 in more than 20% of the cells.").

4. Inappropriate Narrative Voice and Residual Rebuttal Text:

Similar to the issue above, the manuscript contains sentences written in the first-person singular that appear to be copied directly from rebuttal correspondence. Specifically, the phrase "I explicitly state that prior antimicrobial therapy might have suppressed bacterial growth" is highly inappropriate for a multi-authored scientific manuscript and breaks the objective tone of the paper.

Action Required: Please review the entire manuscript to remove any first-person singular pronouns ("I") and defensive/rebuttal phrasing. Revise this specific sentence to an objective scientific statement, such as: "Furthermore, prior antimicrobial therapy might have suppressed bacterial growth."

5. Residual Meta-Text in the Abstract:

The Abstract currently contains administrative details regarding the review process. The Abstract must serve as a standalone summary of the scientific work.

Action Required: Please completely delete the following sentences from the Abstract: "In accordance with the request for methodological transparency, we report that access to the dataset for the specific purposes of this study commenced on 1st June 2024. Formal data analysis specific to the research questions outlined herein began on 1st December 2024 and concluded on 1st March 2025. Statistical analysis was performed on the relevant factors."

6. Phrasing and Typographical Errors in the Methods Timeline:

In the main text (Methods section), the newly added sentences detailing the timeline of data access contain a typographical error and overly bureaucratic phrasing.

Action Required: Please correct the typo ("Data acess") and simplify the syntax. I suggest replacing the current sentences with standard phrasing: "Data access began on June 1, 2024. Data analysis was conducted from December 1, 2024, to March 1, 2025."

7. Severe Readability and Formatting Issues (Text):

Despite previous requests to fix formatting and spacing, the manuscript suffers from a severe lack of readability. The text is poorly structured and riddled with non-standard punctuation.

Inclusion/Exclusion Criteria and Definitions of Infections: These sections are compressed, retain missing spaces ("telephone.Inclusion"), use non-standard punctuation ("1、"), and inexplicably include text left in red font (unaccepted track-changes).

Action Required: A meticulous, line-by-line proofreading of the entire manuscript is mandatory. Remove all non-standard symbols, ensure proper spacing after punctuation marks, use standard line breaks or bullet points for lists, and remove all residual track-change formatting (red text).

8. Inadequate Table Formatting:

Despite previous attention drawn to table presentation, the current tables remain highly difficult to read. The variable name cells are vertically expanded, while the corresponding data values remain vertically center-aligned in their respective cells. This misalignment forces the reader to guess which value belongs to which variable across the row.

Action Required: Please reformat all tables to ensure clear, horizontal alignment (e.g., top-aligning the contents of the cells or strictly ensuring line-by-line correspondence) so that readers can easily and accurately track the data rows.

7. PLOS authors have the option to publish the peer review history of their article (what does this mean?). If published, this will include your full peer review and any attached files.

Reviewer #4: No

Reviewer #5: No

---

## [Author Response · Author response to Decision Letter 5]

8 Mar 2026

Response to Reviewer #5: Thank you very much for your constructive feedback. We have carefully addressed your valuable suggestions as follows:

1. Interpretation of Model Discrimination (AUC):

I commend the addition of the model performance metrics. The calibration of the model appears adequate (Hosmer-Lemeshow p=0.221). However, an interpretative adjustment regarding the discrimination metric (AUC) is required.

The reported AUC is 0.686 (95% CI: 0.514–0.858). In standard biostatistical practice, an AUC between 0.6 and 0.7 is generally classified as "poor" rather than "acceptable". Furthermore, the lower bound of the 95% CI (0.514) approaches 0.50 (random chance). Concluding that the model demonstrates an "acceptable level of both discriminatory ability and goodness-of-fit" overstates its predictive robustness.

Action Required (Results): Please retain the phrase "marginal but statistically significant discriminatory ability," but remove the concluding claim that the discrimination is "acceptable."

Action Required (Limitations): Please explicitly integrate this marginal discrimination into the Limitations section. Add a statement acknowledging that because the AUC is marginal and its lower confidence bound approaches 0.5, the predictive power of the multivariable model is limited, requiring a cautious clinical application of the reported Odds Ratios.

Response :Thank you for your insightful comments regarding the interpretation of our model's discrimination. We agree with your assessment and have revised the manuscript accordingly.

We acknowledge that an AUC of 0.686 (95% CI: 0.514–0.858) falls within the "poor" range per standard biostatistical practice, and that the lower confidence interval approaching 0.5 indicates limited predictive certainty.

In the Results section: We have removed the characterization of the discrimination as "acceptable" and have retained the more precise phrasing describing it as having "marginal but statistically significant discriminatory ability."

In the Limitations section: We have added a explicit statement acknowledging that the model's predictive power is limited due to the marginal AUC and its wide confidence interval. We emphasize that this necessitates cautious interpretation and application of the reported Odds Ratios in a clinical setting.

2. Inadequate Narrative and Missing Methodological Context for Statistical Tests:

The reporting of the multivariate analysis in the Results section reads like raw statistical software output rather than a cohesive scientific narrative. Appending isolated data fragments (e.g., "Hosmer-Lemeshow test: chi^2=10.67, df=8, p=0.221") without proper sentence structure is poor scientific writing. Furthermore, the methodology behind these specific tests must be formally introduced in the Methods section before the results are presented.

Action Required (Methods): Please update the Statistical Analysis section to explicitly state your methodology for evaluating the model (e.g., "Model calibration was assessed using the Hosmer-Lemeshow goodness-of-fit test, and discrimination was evaluated using the Area Under the Receiver Operating Characteristic curve (AUC).")

Action Required (Results): Please rewrite the isolated data fragments into proper, grammatically correct sentences that provide narrative context.

Response :We have made the following revisions:

In the Methods section: Model performance was evaluated using two metrics. First, model calibration—the agreement between observed and predicted outcomes—was assessed using the Hosmer-Lemeshow goodness-of-fit test, where a p＞0.05 indicates adequate calibration. Second, model discrimination—the ability of the model to distinguish between patients with and without the outcome—was evaluated by calculating the Area Under the Receiver Operating Characteristic curve (AUC).

In the Results section: The multivariable logistic regression model demonstrated adequate calibration. The Hosmer-Lemeshow goodness-of-fit test yielded a non-significant result (χ²=10.67, df = 8, p= 0.221), indicating no significant deviation between observed and predicted event rates. Regarding model discrimination, the area under the receiver operating characteristic curve (AUC) was 0.686 (95% CI: 0.514-0.858), suggesting marginal but statistically significant discriminatory ability(Fig 2).

3. Inappropriate Prescriptive Language in Statistical Analysis:

In the Methods section, the sentence "Wherever expected cell frequencies are <5 in more than 20% of cells, the p-values must be recalculated using Fisher’s Exact Test" is stylistically inappropriate. The Methods section must serve as a descriptive record of the analysis actually performed (written in the past tense), rather than prescribing rules or pasting direct instructions from reviewers ("must be recalculated").

Action Required: Please revise this sentence to reflect standard, objective scientific writing (e.g., "Fisher’s exact test was used to compare categorical variables when expected cell frequencies were less than 5 in more than 20% of the cells.").

Response :We have revised the sentence accordingly to reflect standard, objective scientific writing in the past tense. Fisher’s exact test was used to compare categorical variables when expected cell frequencies were less than 5 in more than 20% of the cells.

4. Inappropriate Narrative Voice and Residual Rebuttal Text:

Similar to the issue above, the manuscript contains sentences written in the first-person singular that appear to be copied directly from rebuttal correspondence. Specifically, the phrase "I explicitly state that prior antimicrobial therapy might have suppressed bacterial growth" is highly inappropriate for a multi-authored scientific manuscript and breaks the objective tone of the paper.

Action Required: Please review the entire manuscript to remove any first-person singular pronouns ("I") and defensive/rebuttal phrasing. Revise this specific sentence to an objective scientific statement, such as: "Furthermore, prior antimicrobial therapy might have suppressed bacterial growth."

Response : Modified.

5. Residual Meta-Text in the Abstract:

The Abstract currently contains administrative details regarding the review process. The Abstract must serve as a standalone summary of the scientific work.

Action Required: Please completely delete the following sentences from the Abstract: "In accordance with the request for methodological transparency, we report that access to the dataset for the specific purposes of this study commenced on 1st June 2024. Formal data analysis specific to the research questions outlined herein began on 1st December 2024 and concluded on 1st March 2025. Statistical analysis was performed on the relevant factors."

Response : We have completely removed the specified sentences from the Abstract, as requested.

6. Phrasing and Typographical Errors in the Methods Timeline:

In the main text (Methods section), the newly added sentences detailing the timeline of data access contain a typographical error and overly bureaucratic phrasing.

Action Required: Please correct the typo ("Data acess") and simplify the syntax. I suggest replacing the current sentences with standard phrasing: "Data access began on June 1, 2024. Data analysis was conducted from December 1, 2024, to March 1, 2025."

Response :We have revised the sentences as per your suggestion to ensure clarity and professionalism.

7. Severe Readability and Formatting Issues (Text):

Despite previous requests to fix formatting and spacing, the manuscript suffers from a severe lack of readability. The text is poorly structured and riddled with non-standard punctuation.

Inclusion/Exclusion Criteria and Definitions of Infections: These sections are compressed, retain missing spaces ("telephone.Inclusion"), use non-standard punctuation ("1、"), and inexplicably include text left in red font (unaccepted track-changes).

Action Required: A meticulous, line-by-line proofreading of the entire manuscript is mandatory. Remove all non-standard symbols, ensure proper spacing after punctuation marks, use standard line breaks or bullet points for lists, and remove all residual track-change formatting (red text).

Response :Modified.

Inclusion Criteria:

1.Age < 18 years;

2.ECMO support duration > 48 hours.

Exclusion Criteria:

1.Presence of healthcare-associated infections before ECMO initiation;

2.Patients who required ECMO support before congenital heart surgery.

8. Inadequate Table Formatting:

Despite previous attention drawn to table presentation, the current tables remain highly difficult to read. The variable name cells are vertically expanded, while the corresponding data values remain vertically center-aligned in their respective cells. This misalignment forces the reader to guess which value belongs to which variable across the row.

Action Required: Please reformat all tables to ensure clear, horizontal alignment (e.g., top-aligning the contents of the cells or strictly ensuring line-by-line correspondence) so that readers can easily and accurately track the data rows.

Response :We have completely reformatted all tables to ensure clear horizontal alignment. Specifically, we have implemented top-alignment for all cell contents to guarantee strict line-by-line correspondence between variable names and their associated data values. This revision ensures that readers can easily and accurately track each data row.

---

## [Editor Report · Decision Letter 5]

19 Mar 2026

PONE-D-25-38491R5Risk Factors for Healthcare-associated Infections in Children Undergoing ECMO After Cardiac Surgery for Congenital Heart Disease: A Retrospective StudyPLOS One

Dear Dr. Meng,

Thank you for submitting your manuscript to PLOS ONE. After careful consideration, we feel that it has merit but does not fully meet PLOS ONE’s publication criteria as it currently stands. Therefore, we invite you to submit a revised version of the manuscript that addresses the points raised during the review process.

We look forward to receiving your revised manuscript.

Kind regards,

Giovanni Giordano

Academic Editor

PLOS One

**Journal Requirements:**

**Additional Editor Comments:**

Dear Authors,

Thank you for submitting the revised version of your manuscript entitled

**“Risk Factors for Healthcare-associated Infections in Children Undergoing ECMO After Cardiac Surgery for Congenital Heart Disease: A Retrospective Study.”**

We appreciate the substantial effort you have made in addressing the reviewers’ previous comments. The manuscript has improved in several important aspects, particularly in the clarification of the statistical methodology, the revision of the model performance interpretation, and the overall structure of the Results and Methods sections.

However, after careful evaluation, several critical issues remain that must be addressed before the manuscript can be considered for publication.

Given the multiple rounds of revision already completed, please consider this decision as a **final opportunity to revise the manuscript**. We therefore encourage you to address the following points thoroughly and carefully.

<h3 class="western">**Major Points**</h3>

**1. Interpretation of the multivariable model and causal language**

The current manuscript continues to describe preoperative endotracheal intubation as an “independent risk factor.”

Given the small sample size, wide confidence intervals (OR 4.852, 95% CI 1.174–20.059), and marginal discrimination (AUC 0.686, 95% CI 0.514–0.858), this wording is too strong.

Please revise throughout the manuscript to use more appropriate language, such as:

“associated with increased risk”“associated with the occurrence of HAI”

Causal implications should be avoided.

**2. Confounding by severity (major methodological concern)**

Preoperative intubation is very likely a marker of underlying clinical severity rather than a direct causal factor. This introduces a substantial risk of **confounding by indication**.

This issue is currently underdeveloped in the Discussion.

Please:

Explicitly acknowledge this limitationExpand the discussion on how disease severity may have influenced both exposure (intubation) and outcome (infection)Clarify that the observed association may not reflect a direct causal relationship

**3. Model robustness and statistical limitations**

Although you have improved the reporting of model performance, the limitations are still not sufficiently emphasized.

Please strengthen the Limitations section by clearly stating:

The small sample size and low number of eventsThe borderline events-per-variable ratio (risk of overfitting)The wide confidence intervals indicating imprecisionThe limited discriminatory performance (AUC close to 0.5 at the lower bound)

The model should be presented as **exploratory and hypothesis-generating**, not predictive.

**4. Definition and potential misclassification of infections**

The study includes only microbiologically confirmed infections, while clinically suspected but culture-negative cases were excluded despite receiving treatment.

This approach introduces potential **selection and misclassification bias**, and may underestimate the true incidence of infection.

Additionally, the finding that 100% of infections were respiratory in origin warrants further discussion.

Please:

Clarify this methodological choice more explicitlyDiscuss its implications and limitationsAddress the potential bias introduced by culture-based definitions

**5. Interpretation of ICU length of stay**

The manuscript discusses ICU length of stay as a variable associated with infection; however, this variable is likely influenced by the occurrence of infection itself (reverse causation).

Please:

Clarify the temporal relationshipAvoid implying that ICU stay is a causal factor for infection in this contextRevise the discussion accordingly

**6. Overinterpretation and clinical recommendations**

The Discussion includes extended recommendations (e.g., airway management protocols, VAP prevention strategies) that are not directly supported by the study data.

Please revise this section to:

Focus on findings supported by your resultsAvoid guideline-style or prescriptive recommendations not derived from your analysis

<h3 class="western">**Minor Points**</h3>

The manuscript still contains minor issues with language, punctuation, and formatting (including tables and spacing). A careful, line-by-line proofreading is required.Ensure consistency in terminology (e.g., HAI, ECMO-related infection).Tables should be further improved to enhance readability and alignment.
**Conclusion**

The study addresses a clinically relevant topic and has improved during the revision process. However, the issues outlined above—particularly regarding causal interpretation, confounding, and model limitations—must be fully addressed.

We therefore invite you to submit a **revised version that carefully and comprehensively responds to all points raised above**.

Please note that this will be considered the **final round of revision**, and further decisions will be based on the completeness and quality of your response.

We look forward to receiving your revised manuscript.

Sincerely,

Giovanni Giordano

---

## [Author Response · Author response to Decision Letter 6]

29 Mar 2026

Response to Reviewer Comments：we appreciate the reviewers’ time and expertise in helping us improve this work.

1. Interpretation of the multivariable model and causal language

The current manuscript continues to describe preoperative endotracheal intubation as an “independent risk factor.” Given the small sample size, wide confidence intervals (OR 4.852, 95% CI 1.174–20.059), and marginal discrimination (AUC 0.686, 95% CI 0.514–0.858), this wording is too strong.

Please revise throughout the manuscript to use more appropriate language, such as:

“associated with increased risk”

“associated with the occurrence of HAI”

Causal implications should be avoided.

Response :Thank you for your insightful comment regarding the interpretation of our multivariable model and the use of causal language. We completely agree with your assessment that given the relatively small sample size, the wide confidence intervals (especially for the variable of preoperative endotracheal intubation), and the modest discriminative ability of the model (AUC 0.686), the term "independent risk factor" was overly definitive and potentially misleading.

In response to your suggestion, we have carefully revised the manuscript throughout to temper our language. We have replaced all instances of causal or deterministic phrasing with more appropriate epidemiological terminology that emphasizes association rather than causation.

Specific changes include:

Throughout the manuscript: The term "independent risk factor" has been replaced with "was significantly associated with the occurence of ECMO-related HAI".

In the Results section: We now present the finding as: "preoperative endotracheal intubation remained significantly associated with the occurrence of HAI (OR 4.852, 95% CI 1.174–20.059, P < 0.05)."

In the Discussion section: We have added a note of caution regarding the instability of the estimate due to the wide confidence interval and the modest AUC, suggesting that these findings warrant validation in larger studies.

2 : Confounding by severity (major methodological concern)

Preoperative intubation is very likely a marker of underlying clinical severity rather than a direct causal factor. This introduces a substantial risk of confounding by indication.

This issue is currently underdeveloped in the Discussion.

Please:

Explicitly acknowledge this limitation

Expand the discussion on how disease severity may have influenced both exposure (intubation) and outcome (infection)

Clarify that the observed association may not reflect a direct causal relationship.

Response :We agree that preoperative intubation is likely a marker of underlying clinical severity rather than a direct causal factor. We have substantially revised the Limitations section to: (1) explicitly acknowledge confounding by indication; (2) expand discussion of how disease severity may influence both exposure and outcome; and (3) clarify that the observed association does not necessarily reflect a direct causal relationship.

3:Model robustness and statistical limitations

Although you have improved the reporting of model performance, the limitations are still not sufficiently emphasized.

Please strengthen the Limitations section by clearly stating:

The small sample size and low number of events

The borderline events-per-variable ratio (risk of overfitting)

The wide confidence intervals indicating imprecision

The limited discriminatory performance (AUC close to 0.5 at the lower bound)

The model should be presented as exploratory and hypothesis-generating, not predictive.

Response :We have strengthened the Limitations section to clearly state the small sample size, low number of events, borderline events-per-variable ratio, wide confidence intervals indicating imprecision, and limited discriminatory performance (AUC with lower bound near 0.5). We now explicitly characterize our findings as exploratory and hypothesis-generating.

4: Definition and potential misclassification of infections

The study includes only microbiologically confirmed infections, while clinically suspected but culture-negative cases were excluded despite receiving treatment.

This approach introduces potential selection and misclassification bias, and may underestimate the true incidence of infection.

Additionally, the finding that 100% of infections were respiratory in origin warrants further discussion.

Please:

Clarify this methodological choice more explicitly

Discuss its implications and limitations

Address the potential bias introduced by culture-based definitions

Response :We have clarified in the Methods section that only microbiologically confirmed infections were included, and in the Limitations section we discuss the potential for selection and misclassification bias, underestimation of true infection incidence, and the implications of relying on culture-based definitions. We have also added discussion regarding the predominance of respiratory infections in our cohort.

5 : Interpretation of ICU length of stay

The manuscript discusses ICU length of stay as a variable associated with infection; however, this variable is likely influenced by the occurrence of infection itself (reverse causation).

Please:

Clarify the temporal relationship

Avoid implying that ICU stay is a causal factor for infection in this context

Revise the discussion accordingly

Response :We acknowledge the potential for reverse causation between ICU length of stay and infection. We have revised the Discussion to clarify the temporal limitations and avoid implying that ICU stay is a causal factor for infection.

6 : Overinterpretation and clinical recommendations

The Discussion includes extended recommendations (e.g., airway management protocols, VAP prevention strategies) that are not directly supported by the study data.

Please revise this section to:

Focus on findings supported by your results

Avoid guideline-style or prescriptive recommendations not derived from your analysis

Response :We have deleted the relevant extended recommendations.

7：Minor Points

The manuscript still contains minor issues with language, punctuation, and formatting (including tables and spacing). A careful, line-by-line proofreading is required.

Ensure consistency in terminology (e.g., HAI, ECMO-related infection).

Tables should be further improved to enhance readability and alignment.

Response :We thank the reviewer for the careful reading and constructive suggestions. We have conducted a thorough line-by-line proofreading of the entire manuscript to address issues related to language, punctuation, and formatting. Terminology has been standardized throughout the text; "healthcare-associated infection (HAI)" and "ECMO-related HAI" are now used consistently. In addition, all tables have been reformatted to improve readability and alignment. We believe these revisions have substantially enhanced the overall quality and clarity of the manuscript.

---

## [Editor Report · Decision Letter 6]

13 Apr 2026

Risk Factors for Healthcare-associated Infections in Children Undergoing ECMO After Cardiac Surgery for Congenital Heart Disease: A Retrospective Study

PONE-D-25-38491R6

Dear Dr. Meng,

We’re pleased to inform you that your manuscript has been judged scientifically suitable for publication and will be formally accepted for publication once it meets all outstanding technical requirements.

Kind regards,

Giovanni Giordano

Academic Editor

PLOS One
---

## [Editor Report · Acceptance letter]

PONE-D-25-38491R6

PLOS One

Dear Dr. Meng,

I'm pleased to inform you that your manuscript has been deemed suitable for publication in PLOS One. Congratulations! Your manuscript is now being handed over to our production team.

Kind regards,

on behalf of

Dr. Giovanni Giordano

Academic Editor

PLOS One